# Molecular Evolution of Attachment Glycoprotein (G) and Fusion Protein (F) Genes of Respiratory Syncytial Virus ON1 and BA9 Strains in Xiamen, China

Yong-Peng Sun,[a] Si-Yu Lei,[a] Ying-Bin Wang,[a] Yi-Zhen Wang,[a] Hong-Sheng Qiang,[a] Yi-Fan Yin,[a] Ze-Min Jiang,[a] Min Zhu,[b] Xiao-Li Chen,[b] Hui-Ming Ye,[b] Zi-Zheng Zheng,[a] Ning-Shao Xia[a]

[a]State Key Laboratory of Molecular Vaccinology and Molecular Diagnostics, National Institute of Diagnostics and Vaccine Development in Infectious Diseases, School of Public Health, Xiamen University, Xiamen, Fujian, People's Republic of China
[b]Department of Clinical Laboratory, Women and Children's Hospital, School of Medicine, Xiamen University, Xiamen, Fujian, People's Republic of China

Yong-Peng Sun, Si-Yu Lei, and Ying-Bin Wang contributed equally to this article. Author order was determined by author's contribution to work and the corresponding author after negotiation.

**ABSTRACT** Monitoring viral transmission and analyzing the genetic diversity of a virus are imperative to better understand its evolutionary history and the mechanism driving its evolution and spread. Especially, effective monitoring of key antigenic mutations and immune escape variants caused by these mutations has great scientific importance. Thus, to further understand the molecular evolutionary dynamics of respiratory syncytial virus (RSV) circulating in China, we analyzed nasopharyngeal swab specimens derived from hospitalized children ≤5 years old with acute respiratory tract infections (ARIs) in Xiamen during 2016 to 2019. We found that infants under 6 months of age (52.0%) were the main population with RSV infection. The prevalent pattern "BBAA" of RSV was observed during the epidemic seasons. RSV ON1 and BA9 genotypes were the dominant circulating strains in Xiamen. Interestingly, we observed four Xiamen-specific amino acid substitution combinations in the G protein and several amino acid mutations primarily occurring at antigenic sites Ø and V in the F protein. Our analyses suggest that introduction of new viruses and local evolution are shaping the diversification of RSV strains in Xiamen. This study provides new insights on the evolution and spread of the ON1 and BA9 genotypes at local and global scales.

**IMPORTANCE** Monitoring the amino acid diversity of the RSV G and F genes helps us to find the novel genotypes, key antigenic mutations affecting antigenicity, or neutralizing antibody-resistant variants produced by natural evolution. In this study, we analyzed the molecular evolution of G and F genes from RSV strains circulating in Xiamen, China. These data provide new insights on local and global transmission and could inform the development of control measures for RSV infections.

**KEYWORDS** RSV, hospitalized children, attachment glycoprotein, fusion protein, molecular epidemiology

Respiratory syncytial virus (RSV) is the leading pathogen causing acute respiratory tract infections (ARIs) among children ≤5 years old (1). RSV F and G glycoproteins are the main target antigens of neutralizing antibody and vaccine development. Based on the G gene, RSV is divided into groups A and B (2). To date, according to the sequence variations of the second hypervariable region (HVR2) of the G protein, RSV A strains have been categorized into 14 genotypes (GA1 to -7, SAA1, CB-A, NA1 to -4, and ON1) (3–7), and RSV B strains have been classified into 26 genotypes (GB1 to -4, SAB1 to -4, URU1 and -2, CB1, BA1 to -14, and BAC) (3, 5, 6, 8–12). The ON1 genotype, initially found in Ontario, Canada, in 2010, was characterized by a 24-amino-acid (aa) duplication in the G gene HVR2 (7). The BA9 genotype was first identified in Niigata,

Address correspondence to Hui-Ming Ye, yehuiming@xmu.edu.cn, or Zi-Zheng Zheng, zhengzizheng@xmu.edu.cn.

The authors declare no conflict of interest.

**TABLE 1** Demographic characteristics of the study subjects

| Characteristic | No. (%) | | P |
| --- | --- | --- | --- |
| | RSV A (n = 321) | RSV B (n = 350) | |
| Age (mo) | | | 0.144 |
| ≤6 | 177 (50.7) | 172 (49.3) | |
| 6–12 | 55 (39.9) | 83 (60.1) | |
| 12–24 | 47 (45.6) | 56 (54.4) | |
| 24–60 | 42 (51.9) | 39 (48.1) | |
| Sex | | | 0.406 |
| Male | 202 (46.7) | 231 (53.3) | |
| Female | 119 (50.0) | 119 (50.0) | |
| Yr of admission | | | <0.001 |
| 2016 | 52 (26.3) | 146 (73.7) | |
| 2017 | 71 (35.0) | 132 (65.0) | |
| 2018 | 144 (70.2) | 61 (29.8) | |
| 2019 | 54 (83.1) | 11 (16.9) | |

Japan, in 2006 (12). Like other BA genotypes, a sequence repeat of 20 amino acids was also inserted into HVR2 (13). The inserted sequence duplication is a key antigenic region including multiple epitopes, which could be responsible for the G protein antigenicity and could augment the attachment ability and adaptability of RSV (14). ON1 and BA genotype circulation is usually accompanied by length changes and substitutions in the amino acid sequence of HVR2. At present, both genotypes have become entirely the dominant RSV strains worldwide. The local and global continued surveillance of RSV is crucial to gain a better understanding of the evolutionary and epidemiological dynamics of viruses and to facilitate the evaluation of the underlying effect of genetic polymorphisms on RSV vaccines and prophylactic drug development. Here, we investigated the molecular evolutionary dynamics of the G and F genes of RSV strains circulating in Xiamen, China, from March 2016 to April 2019.

## RESULTS

**Demographic characteristics of the study subjects.** A total of 1,026 nasopharyngeal swab specimens collected from March 2016 to April 2019 were originally tested as RSV positive by direct immunofluorescence, 671 of which were ultimately identified as positive by one-step quantitative real-time PCR (qRT-PCR). A total of 321 (47.8%) were RSV A infections, and 350 (52.2%) were RSV B infections. The age of hospitalized children ranged from 1 day to 5 years. Infants ≤6 months of age (52.0%) were the main population with RSV infection. The male-to-female ratio was 1.82 (433 boys and 238 girls) (Table 1). Although both RSV A and B strains cocirculated in the 2016-to-2019 RSV seasons, a prevalent pattern, "BBAA," of the dominant strain in Xiamen was observed in this period, which meant that the predominant group switched at a certain point in time during 2017 to 2018.

**Time-scaled phylogenetic analysis of RSV G protein by MCMC method.** A total of 228 G protein ectodomain sequences (182 RSV A and 46 RSV B) were successfully achieved. To estimate the dynamics of the nucleotide substitutions of G protein HVR2 of Xiamen strains in this study, all Xiamen strains and reference genotypes from GenBank were used to construct time-scaled phylogenetic trees by the Markov chain Monte Carlo (MCMC) method. Phylogenetic analysis illustrated that all RSV A strains were of the ON1 genotype and that all RSV B strains were of the BA9 genotype (Fig. 1A and B). The time-scaled maximum clade credibility (MCC) trees showed that the time to the most recent common ancestor (tMRCA) was assessed as the length of time since approximately 1932 (95% highest probability density [95% HPD]: 1884 to 1955) for RSV A and since approximately 2007 (95% HPD: 2002 to 2010) for ON1, which was in line with previous results (15, 16). Furthermore, the tMRCA for RSV B was assessed as the

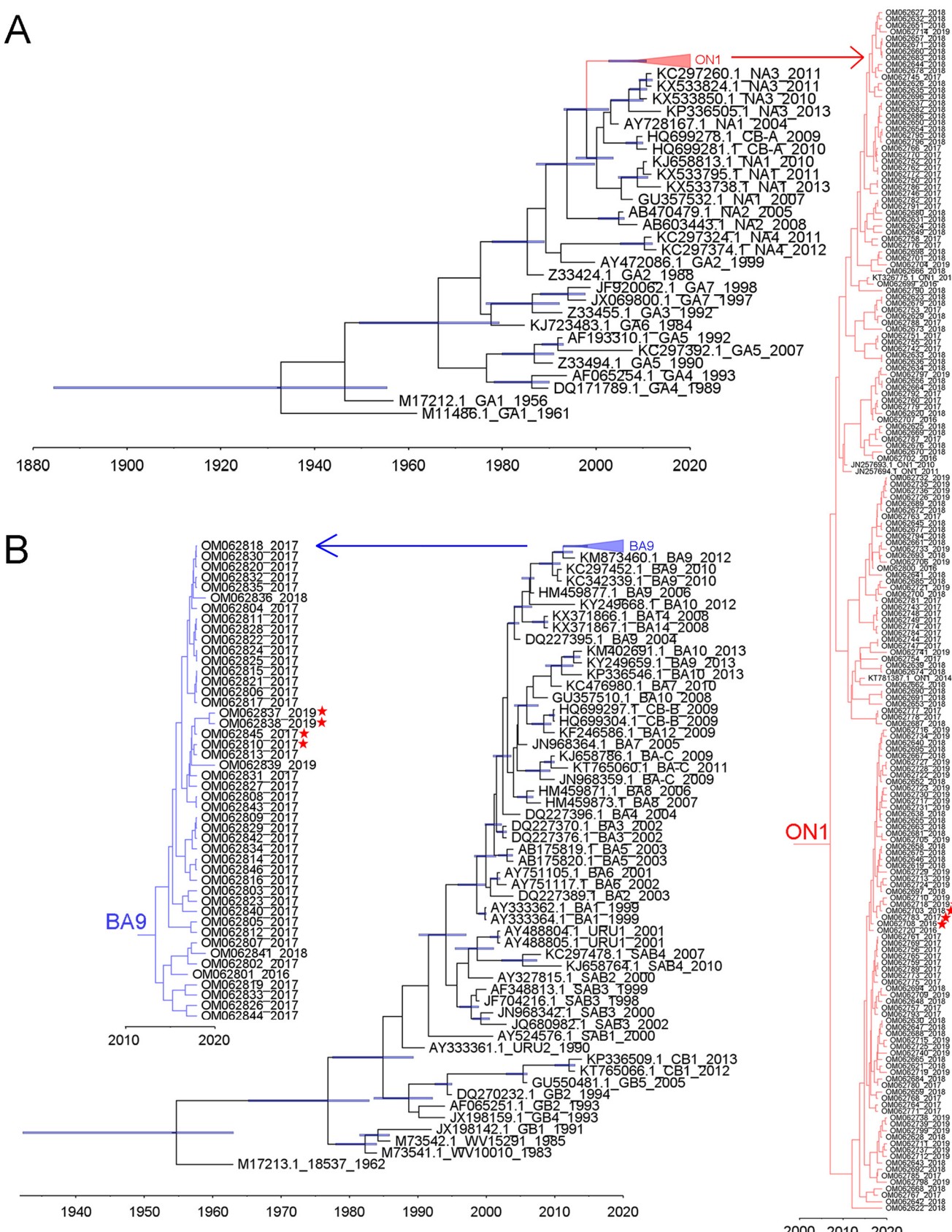

**FIG 1** Time-scaled MCC trees for G protein of Xiamen RSV A and B strains and reference strains representing known genotypes. The branches of Xiamen ON1 (A) and BA9 (B) are colored red and blue, respectively, and then shown individually alongside in the graph. Four representative sequences from Xiamen ON1 and BA9 strains collected from successive epidemic seasons are labeled by red stars. Scale bars represent the unit of time (year).

length of time since approximately 1954 (95% HPD: 1932 to 1963) and that for BA9 as since approximately 2009 (95% HPD: 2007 to 2010), consistent with previous reports (15, 16). The mean evolutionary rate of the Xiamen ON1 genotype was $1.95 \times 10^{-3}$ substitutions/site/year (95% HPD: $1.45 \times 10^{-3}$ to $2.48 \times 10^{-3}$), which was lower than that of the Xiamen BA9 genotype ($3.15 \times 10^{-3}$ substitutions/site/year; 95% HPD: $2.45 \times 10^{-4}$ to $3.95 \times 10^{-3}$). In addition, we observed that one clade consisted of closely related strains collected from successive epidemic seasons (2016 to 2019) in the ON1 lineage in the phylogenetic tree for RSV A strains. Four representative sequences were labeled by red stars. Likewise, in the BA9 lineage in the phylogenetic tree for RSV B strains, we observed that Xiamen BA9 strains circulating in 2017 occurred again in 2019. The representative sequences were also marked by red stars. The time-scaled MCC trees show local transmission and persistence through consecutive RSV seasons.

**Subgenotype definition and polymorphism analysis of RSV G protein.** To define the subgenotypes for Xiamen ON1 and BA9 genotypes and lineages within subgenotypes, maximum-likelihood (ML) trees of Xiamen ON1 and BA9 strains were constructed by IQ-TREE. Based on the statistical support of ultrafast bootstrap values of ≥80%, we set a cutoff 0.01 patristic distance to define subgenotype for ON1 genotype and 0.005 for BA9 genotype. All ON1 strains were classified into 2 subgenotypes (ON1.1 and ON1.2). Four lineages (ON1.2.1, ON1.2.2, ON1.2.3, and ON1.2.4) were further identified within the ON1.2 subgenotype (Fig. 2A). The whole BA9 strains were also classified into 2 subgenotypes (BA9.1 and BA9.2) (Fig. 2B).

The G protein ectodomain consists of two hypervariable regions (HVR1 and HVR2), separated by a central conserved domain (CCD) and a highly basic heparin-binding domain (HBD) (Fig. 3A). The amino acid polymorphisms of G protein ectodomain can more fully reflect the evolutionary dynamics of G protein than two hypervariable regions. Based on the subgenotype and lineage definition in Fig. 2A and B, we investigated amino acid polymorphisms of the extracellular domain (aa 68 to 320 for RSV A; aa 68 to 310 for RSV B) of the RSV G protein, and the representative sequences from each lineage, marked by red stars in Fig. 2A and B, were aligned with the reference strains for ON1 (GenBank accession no. JN257693) and BA9 (GenBank accession no. DQ227395). Color shading highlights the amino acid variations of different lineages. Alignment of deduced amino acid sequences illustrated that each lineage of Xiamen ON1 and BA9 strains shared characteristic substitution combinations (Fig. 3B and C). Notably, we found four novel mutation combinations, T113I/V131D/N178G/H258Q/H266L in the ON1.1 subgenotype and V225A/G232R/E263V/T320A in the ON1.2.4 lineage in the phylogenetic tree of RSV A G and D213Y/L217P/A269V/L284P/A301P in the BA9.1 lineage and A131T/T137I/T288I/T310I in the BA9.2 lineage in the phylogenetic tree of RSV B G.

We next analyzed the amino acid mutation frequency in the extracellular domain of Xiamen RSV G relative to the ON1 and BA9 references (Fig. 4A). Overall, the G protein HVR2 of RSV A exhibited higher sequence diversity than that of RSV B. Actually, we also calculated the percentage of each characteristic mutation combination and observed that the ON1.1 subgenotype with T113I/V131D/N178G/H258Q/H266L (41.2%, 75/182) and the BA9.2 subgenotype with A131T/T137I/T288I/T310I (47.8%, 22/46) were the predominant strains in Xiamen during the 2016-to-2019 RSV seasons, followed by the ON1.2.1 subgenotype with K134I/T249I/E262K (19.8%, 36/182) and the ON1.2.4 subgenotype with V225A/G232R/E263V/T320A (8.8%, 16/182) (Fig. 4A).

The characteristic sequence repeats were present in G protein HVR2 of ON1 and BA strains with either 23 aa (aa 261 to 283 and 285 to 307 for ON1) or 20 aa (aa 240 to 259 and 260 to 279 for BA), which had several highly conserved sequence motifs considered indicators to evaluate genetic diversity and viral evolution. The amino acid sequence "G**YL**SPSQ" coexisted in the first and second duplicated regions of the ON1 genotype (Fig. 4B). Half of the Xiamen strains had the same signature motif "YL-YL" in both repeated regions, followed by the other motifs "YP-YP" (11.0%), "HP-YL" (4.9%), "YP-YL" (3.8%), "YL-HP" (2.7%), and "YL-YP" (1.1%). Likewise, the G gene of the BA9

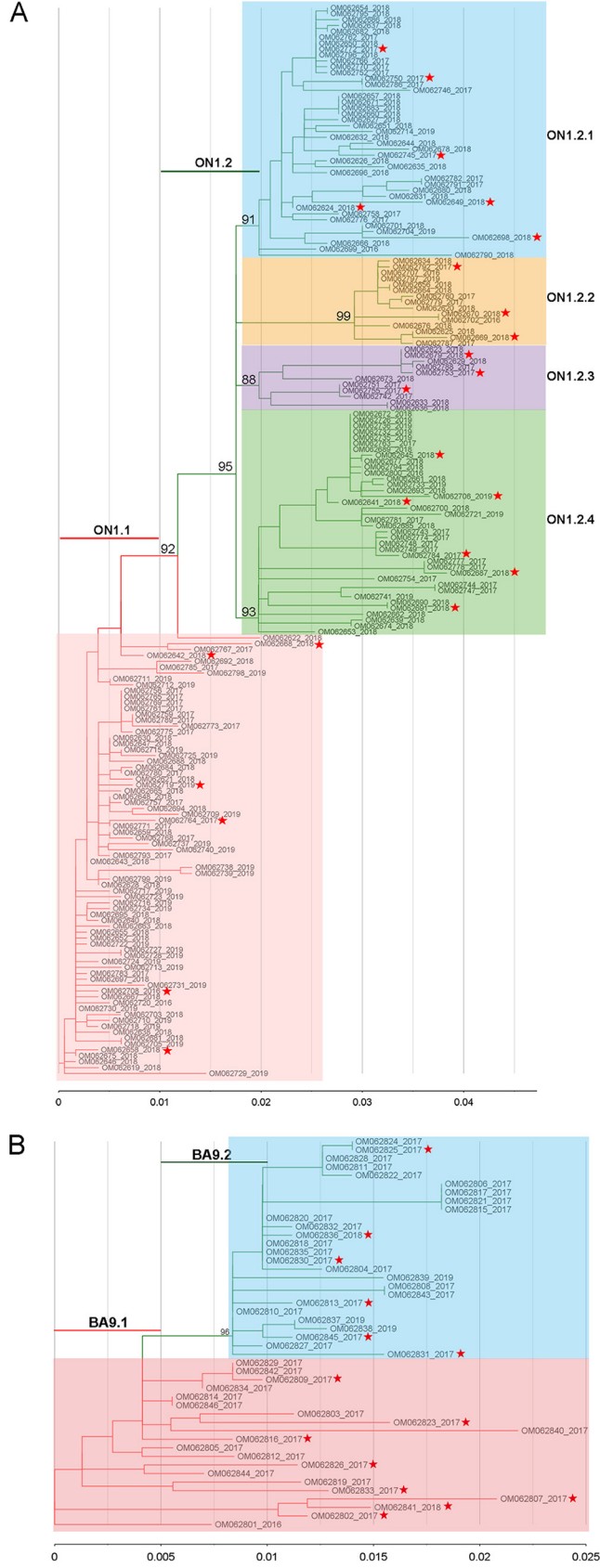

**FIG 2** ML trees of Xiamen ON1 (A) and BA9 (B) strains were constructed by IQ-TREE. Branches of different subgenotypes are highlighted by different colors. Each subgenotype name is listed above the colored line. Each lineage is marked by color shading. Scale bar shows patristic distance by ranges of 0.01 for RSV A and 0.005 for RSV B. Ultrafast bootstrap values of each ancestral lineage node as statistical support are displayed on the trees.

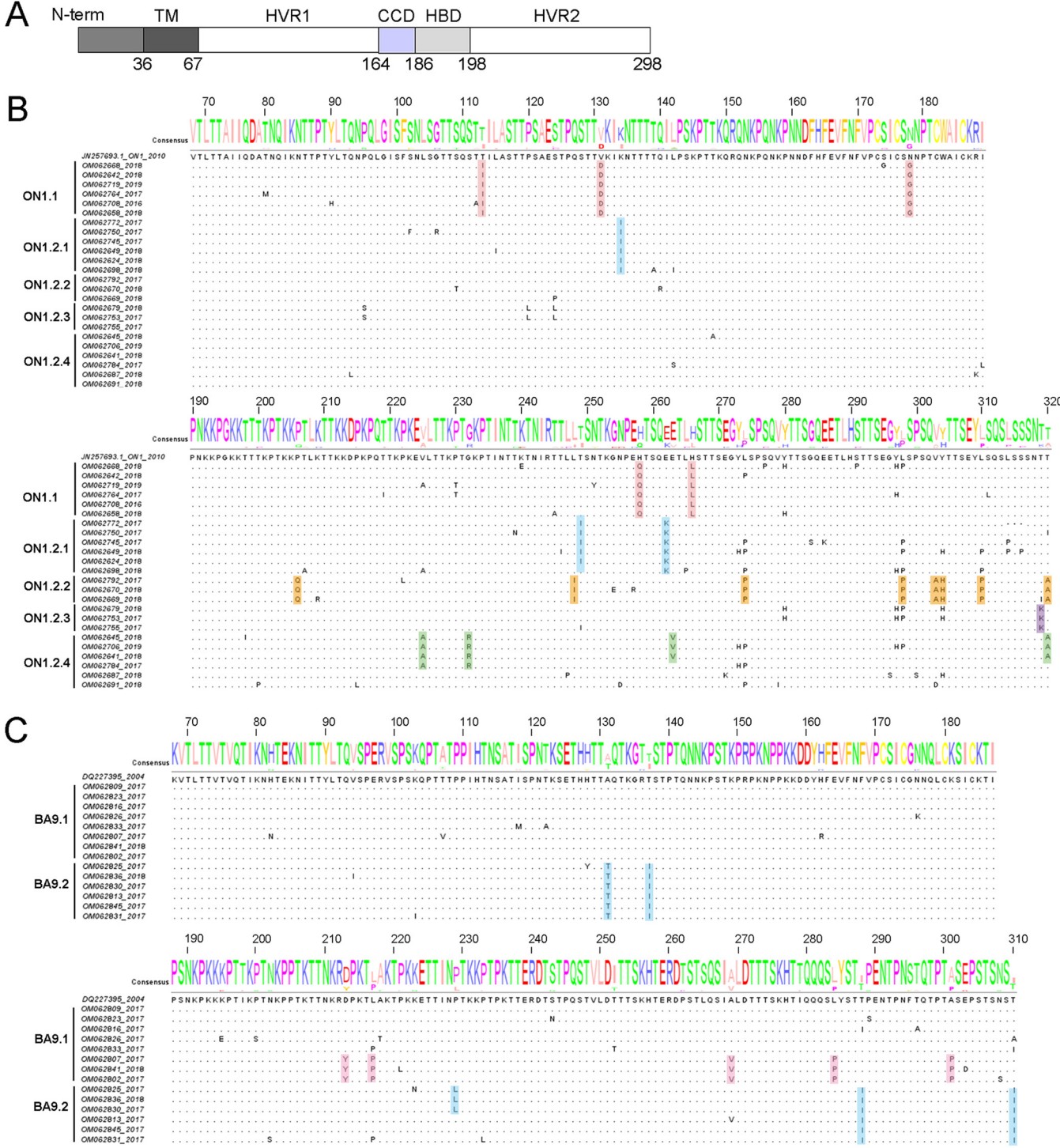

**FIG 3** Characteristic substitution combinations of the G protein ectodomain of Xiamen ON1 and BA9 sublineages. (A) Schematic of the RSV G protein. The RSV A2 G protein (298 amino acids) includes cytoplasmic and transmembrane regions (TM), two hypervariable regions (HVR1 and HVR2), a central conserved domain (CCD), and a highly basic heparin-binding domain (HBD). (B and C) Alignment of deduced amino acid sequences of the G protein ectodomain of Xiamen ON1 (aa 68 to 320) and BA9 (aa 68 to 310) strains relative to the reference strains ON1 (JN257693) and BA9 (DQ227395). The representative sequences marked by red stars from each lineage were aligned with the reference strains for ON1 and BA9 according to Fig. 2A and B. Color shading highlights the unique amino acid variations of different lineages.

genotype included conserved amino acid sequences in both repeated regions, "ST**V**LDTT" and "SI**A**LDTT" (Fig. 4B). "TV-IA" (84.8%) was the predominant motif, followed by the other motif variations "TV-IV" (10.9%), "TV-TA" (2.2%), and "TV-IT" (2.2%). The results above demonstrated that the G protein HVR2 of multiple strains currently

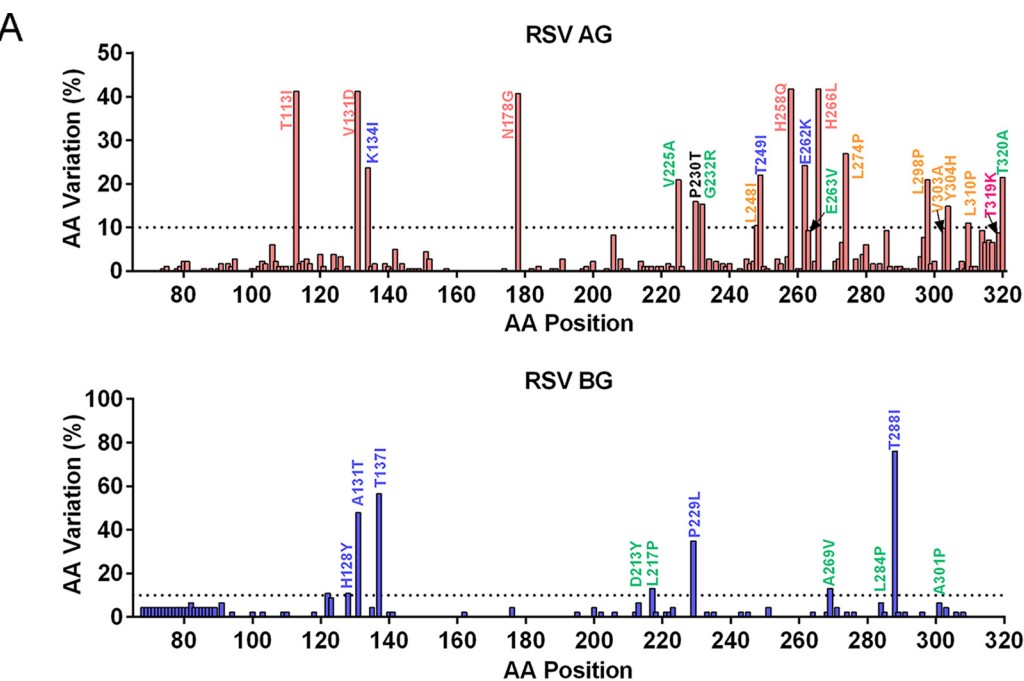

**B**

**RSV A**

| Cluster | % | # | | Duplication # 1 | | | | | | | | | | | | | | | | | | | | | | | | Duplication # 2 | | | | | | | | | | | | | | | | | | | | | |
|---|---|---|---|---|---|---|---|---|---|---|---|---|---|---|---|---|---|---|---|---|---|---|---|---|---|---|---|---|---|---|---|---|---|---|---|---|---|---|---|---|---|---|---|---|---|---|---|
| | | AA position | 261 | 263 | 265 | 267 | 269 | 271 | 273 | 275 | 277 | 279 | 281 | 283 | | | | | | | | | | | | | 285 | 287 | 289 | 291 | 293 | 295 | 297 | 299 | 301 | 303 | 305 | 307 | | | | | | | | | | | | |
| | | | Q E E T L H S T T S E G Y L S P S Q V Y T T S | | | | | | | | | | | | | | | | | | | | | | | Q E E T L H S T T S E G Y L S P S Q V Y T T S | | | | | | | | | | | | | | | | | | | | | | | |
| | # | variation | 1 44 17 0 4 76 0 0 0 4 5 12 49 0 0 5 0 7 11 0 3 0 | | | | | | | | | | | | | | | | | | | | | | | 0 17 2 0 2 2 1 1 0 1 0 6 14 38 3 4 0 0 18 27 0 0 1 | | | | | | | | | | | | | | | | | | | | | | | |
| | 26.9% | 49 | L (267) · Y L (273/274) | | | | | | | | | | | | | | | | | | | | | | | Y L (297/298) | | | | | | | | | | | | | | | | | | | | | | | |
| | 14.3% | 26 | Y L (273/274) | | | | | | | | | | | | | | | | | | | | | | | Y L (297/298) | | | | | | | | | | | | | | | | | | | | | | | |
| | 9.3% | 17 | K (263) · Y L | | | | | | | | | | | | | | | | | | | | | | | Y L | | | | | | | | | | | | | | | | | | | | | | | |
| YL-YL | 6.6% | 12 | V (265) · Y L | | | | | | | | | | | | | | | | | | | | | | | Y L | | | | | | | | | | | | | | | | | | | | | | | |
| (54.9%) | 4.4% | 8 | K (263) · Y L | | | | | | | | | | | | | | | | | | | | | | | K (293) · Y L | | | | | | | | | | | | | | | | | | | | | | | |
| | 1.6% | 3 | Y L · I (281) | | | | | | | | | | | | | | | | | | | | | | | Y L | | | | | | | | | | | | | | | | | | | | | | | |
| | 1.1% | 2 | K (263) · P (267) · Y L | | | | | | | | | | | | | | | | | | | | | | | Y L | | | | | | | | | | | | | | | | | | | | | | | |
| | 1.1% | 2 | L (269) · Y L | | | | | | | | | | | | | | | | | | | | | | | Y (293) · Y L | | | | | | | | | | | | | | | | | | | | | | | |
| | 1.1% | 2 | K (271) · Y L | | | | | | | | | | | | | | | | | | | | | | | S (295) Y L S (299) | | | | | | | | | | | | | | | | | | | | | | | |
| YP-YP | 8.2% | 15 | Y P (273/274) | | | | | | | | | | | | | | | | | | | | | | | Y P · A H (305/307) | | | | | | | | | | | | | | | | | | | | | | | |
| (11.0%) | 1.6% | 3 | L (267) · Y P | | | | | | | | | | | | | | | | | | | | | | | Y P | | | | | | | | | | | | | | | | | | | | | | | |
| | 1.1% | 2 | K (263) · Y P | | | | | | | | | | | | | | | | | | | | | | | K (293) · Y P | | | | | | | | | | | | | | | | | | | | | | | |
| HP-YL | 2.7% | 5 | H P (273/274) | | | | | | | | | | | | | | | | | | | | | | | Y L | | | | | | | | | | | | | | | | | | | | | | | |
| (4.9%) | 1.1% | 2 | K · S H P · I (281) | | | | | | | | | | | | | | | | | | | | | | | K (293) · Y L | | | | | | | | | | | | | | | | | | | | | | | |
| | 1.1% | 2 | K · S H P | | | | | | | | | | | | | | | | | | | | | | | Y L | | | | | | | | | | | | | | | | | | | | | | | |
| YP-YL | 1.6% | 3 | L (267) · Y P | | | | | | | | | | | | | | | | | | | | | | | Y L | | | | | | | | | | | | | | | | | | | | | | | |
| (3.8%) | 1.1% | 2 | V (265) · Y P | | | | | | | | | | | | | | | | | | | | | | | Y L | | | | | | | | | | | | | | | | | | | | | | | |
| | 1.1% | 2 | Y P · I (277) | | | | | | | | | | | | | | | | | | | | | | | Y L · D (303) | | | | | | | | | | | | | | | | | | | | | | | |
| YL-HP | 2.7% | 5 | Y L | | | | | | | | | | | | | | | | | | | | | | | H P · H | | | | | | | | | | | | | | | | | | | | | | | |
| YL-YP | 1.1% | 2 | K · Y L | | | | | | | | | | | | | | | | | | | | | | | Y P | | | | | | | | | | | | | | | | | | | | | | | |
| MINOR | <1.0% | 18 | H G F · P · H K | | | | | | | | | | | | | | | | | | | | | | | D · F · P A · D · R L · I · P | | | | | | | | | | | | | | | | | | | | | | | |
| | each | 1 | T | | | | | | | | | | | | | | | | | | | | | | | I | | | | | | | | | | | | | | | | | | | | | | | |

**RSV B**

| Cluster | % | # | | Duplication # 1 | | | | | | | | | | Duplication # 2 | | | | | | | | | |
|---|---|---|---|---|---|---|---|---|---|---|---|---|---|---|---|---|---|---|---|---|---|---|---|
| | | AA position | 240 | 242 | 244 | 246 | 248 | 250 | 252 | 254 | 256 | 258 | 260 | 262 | 264 | 266 | 268 | 270 | 272 | 274 | 276 | 278 |
| | | | T E R D T S T S Q S T V L D T T T S K H | | | | | | | | | | T E R D T S T S Q S I A L D T T T S K H | | | | | | | | | |
| | # | variation | 0 0 0 0 0 1 0 46 0 0 0 0 0 2 44 0 0 0 0 0 | | | | | | | | | | 0 0 0 0 0 0 1 1 0 0 1 6 0 2 0 0 1 0 1 0 | | | | | | | | | |
| | 69.6% | 32 | P (248) · T V (250/251) · I (254) | | | | | | | | | | I A (268/270) | | | | | | | | | |
| | 4.3% | 2 | P · T V · I | | | | | | | | | | I A · N (272) | | | | | | | | | |
| TV-IA | 2.2% | 1 | P · T V · N I | | | | | | | | | | I A | | | | | | | | | |
| (84.8%) | 2.2% | 1 | N (246) · P · T V · I | | | | | | | | | | I A | | | | | | | | | |
| | 2.2% | 1 | P · T V · I | | | | | | | | | | I A · I (274) | | | | | | | | | |
| | 2.2% | 1 | Q (248) · T V · I | | | | | | | | | | I A | | | | | | | | | |
| | 2.2% | 1 | P · T V | | | | | | | | | | I A | | | | | | | | | |
| TV-IV | 8.7% | 4 | P · T V · I | | | | | | | | | | I V | | | | | | | | | |
| (10.9%) | 2.2% | 1 | P · T V · N I | | | | | | | | | | I V | | | | | | | | | |
| TV-TA | 2.2% | 1 | P | | | | | | | | | | A P (266/268) · T A (272/274) · E (278) | | | | | | | | | |
| TV-IT | 2.2% | 1 | P · T V · I | | | | | | | | | | I T (268/270) | | | | | | | | | |

**FIG 4** Polymorphism analysis of the G protein ectodomain of Xiamen ON1 and BA9 strains. (A) Plots of amino acid variation frequency by position of the G protein ectodomain (A, aa 68 to 320; B, aa 68 to 310) to the reference strains ON1 (JN257693)

circulating in Xiamen is evolving continuously owing to immune pressure. Local RSV ON1 and BA9 strains have evolved with novel mutation combinations in Xiamen and become predominant strains rapidly spreading currently.

**Phylogenetic analysis of RSV G protein.** To investigate the genetic relationship between Xiamen strains and global circulating strains in recent years, we constructed ML trees of Xiamen strains with 408 RSV group A sequences and 696 RSV group B sequences circulating worldwide during 2010 to 2019 downloaded from GenBank. We set a cutoff 0.005 patristic distance to define lineages for RSV A and a 0.01 distance for RSV B. All RSV A strains were clustered into 3 lineages (see Fig. S1A in the supplemental material). Xiamen ON1 strains (marked by colored dots) were distributed in all 3 lineages (Fig. 5A). However, RSV B strains were clustered into 4 lineages (Fig. S1B). All the Xiamen BA9 strains (marked by colored dots) belonged to lineage 4 (Fig. 5B). Xiamen ON1 and BA9 strains were not independently clustered into a clade in the RSV A and B G trees. Each local cluster contained not only Xiamen ON1 or BA9 strains but also strains from other provinces of China and/or other countries.

The specific substitution combination T113I/V131D/N178G/H258Q/H266L had been reported by Li et al. in Guangdong, China, in 2020, marked by red stars in lineage 3 (17). This substitution combination was also found in Russia 2019 strains (MT422269 to MT422271, labeled by blue stars) and United States 2018–2019 strains (MN306017, MN306029, MN306045, and MN306048, labeled by green stars) based on Fig. 5A. However, our results suggested that this substitution combination occurred as early as 2016 in Xiamen (Fig. 1 and 2A), which at present was the earliest time this substitution combination was found. Although A131T/T137I/T288I/T310I was found in Russia 2019 strains (MZ151851 and MT373703 to MT373705, orange stars), Australia 2017-to-2019 strains (MW020595 and MW160815, purple stars), a Switzerland 2019 strain (MT107528, a green star), and a Japan 2018 strain (LC495297, a red star) based on Fig. 5B, interestingly, the P229L substitution also occurred in partial BA9.2 subgenotype strains (Fig. 3C). The novel substitution combinations V225A/G232R/E263V/T320A, A131T/T137I/T288I/T310I, and D213Y/L217P/A269V/L284P/A301P were first reported by our group. Additionally, the K134I/T249I/E262K substitution combination in the ON1.2.1 lineage of RSV A was detected in Changchun, Shandong, and Beijing, China, during the 2014–2015 RSV seasons (18).

**Phylogenetic analysis of RSV F protein by the ML method.** A total of 173 F gene sequences (118 RSV A and 55 RSV B) were ultimately obtained in this study. We next investigated the genetic relationship between Xiamen strains and strains circulating recently worldwide. Phylogenetic trees of the RSV F protein were constructed by the ML method with 829 RSV group A sequences and 629 RSV group B sequences circulating globally during 2010 to 2019 derived from GenBank (Fig. S2). Based on the statistical support of ultrafast bootstrap values of ≥80%, a cutoff 0.0025 patristic distance was set to define lineages for RSV A F and an 0.01 distance was used for RSV B F. All RSV A F sequences were clustered into 2 lineages. The F gene of Xiamen strains was distributed in both of lineages 1 and 2 (Fig. S2A). Significantly, partial Xiamen RSV A strains evolved into a new branch in lineage 2 as well as 2018–2019 American (red stars) and Russian (blue stars) strains (Fig. S3A). All RSV B F sequences were separated into 3 lineages. All Xiamen strains were contained in lineage 3 (Fig. S2B). Some Xiamen RSV B strains evolved into a new clade in lineage 3 with 2017-to-2019 Australian (purple stars), Japanese (cyan stars), American (yellow stars), Switzerland (green stars), Russian (orange stars), and Chinese (red stars) strains (Fig. S3B).

**Amino acid polymorphisms in the F protein of Xiamen strains.** Amino acid mutation analysis of Xiamen RSV A and B F protein was performed in comparison with the two reference sequences, RSV A2 strain (GenBank accession no. KT992094) and RSV

**FIG 4** Legend (Continued)
and BA9 (DQ227395). Red, RSV A; blue, RSV B. Compared with the references, variations at a frequency of >10% (dashed lines) are labeled. (B) Sequence alignment of the sequence repeats in the G protein HRV2 of the ON1 and BA9 strains. The sequence repeats (aa 261 to 283 and 285 to 307 for RSV A; aa 240 to 259 and 260 to 279 for RSV B) are labeled duplication #1 and duplication #2. Motifs used for the analysis are marked by gray shading. Clustering was performed according to the sequences in the duplicated regions. Blank spaces indicate similar amino acids to the references. The percentages of each representative cluster for the ON1 and BA9 strains are listed in the table.

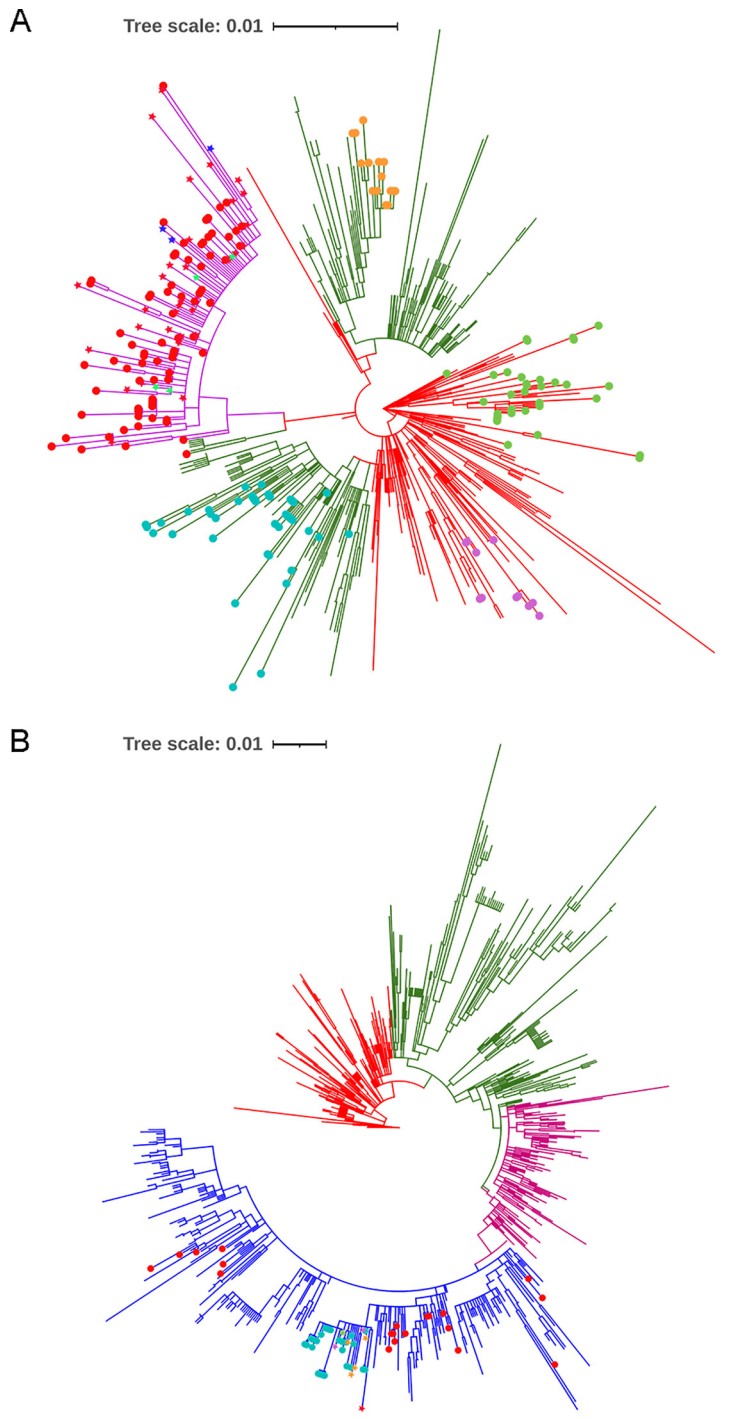

**FIG 5** ML trees for G protein of Xiamen RSV A and B strains and global circulating strains (2010 to 2019). ML trees for RSV A (A) and B (B) strains were constructed using the ML method with IQ-TREE and displayed in circular format. Branches of different lineages are highlighted by different colors (red for lineage 1, green for lineage 2, purple for lineage 3, and blue for lineage 4). Xiamen strains obtained in this study are marked by colored dots. Guangdong, Russia, and America strains are labeled by red, blue, and green stars, respectively, in lineage 3 in the ML tree for RSV A strains. Similarly, Switzerland (a green star), Japan (a red star), Australia (purple stars), and Russia (orange stars) strains are labeled by colored stars in lineage 4 in the ML tree for RSV B strains.

9320 strain (GenBank accession no. AY353550). Out of 549 amino acid residues in the F protein without the leader sequence, 41 (7.5%) amino acid sites were mutated in RSV A F and 32 (5.8%) in RSV B F (Fig. 6). Overall, the Xiamen RSV F gene showed higher conservation in amino acid positions, consistent with previous views (19).

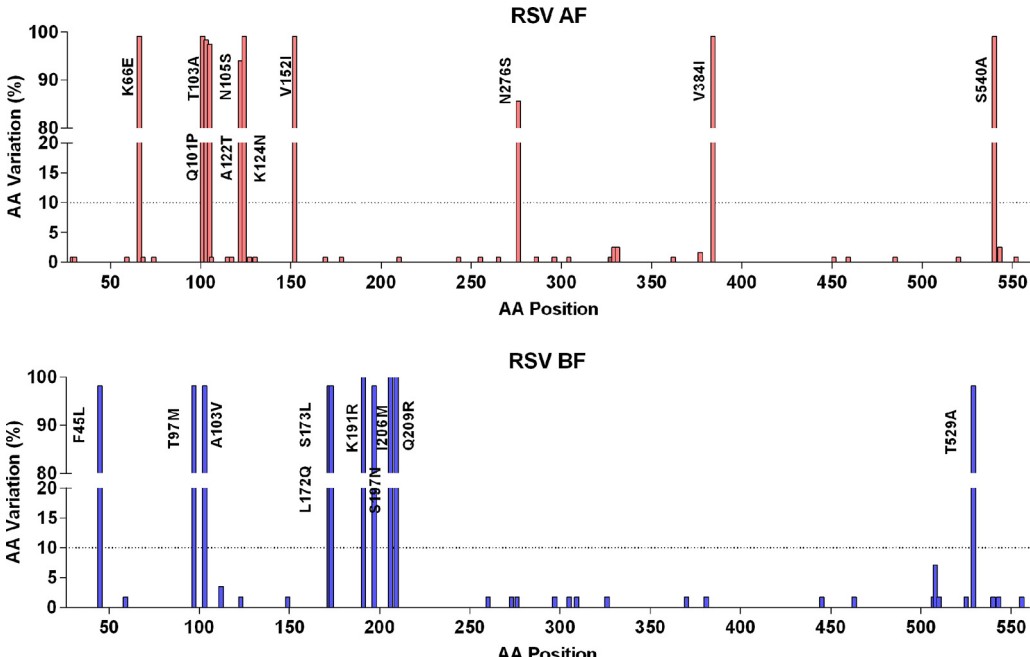

**FIG 6** Polymorphisms in the F protein of Xiamen RSV A and B strains. Plots of amino acid variation frequency by position of the F protein (aa 26 to 574) to the reference sequences RSV A2 (KT992094) and 9320 (AY353550). Red, RSV A; blue, RSV B. Compared with the references, variations at a frequency of >10% (dashed lines) are labeled.

At present, seven antigenic sites (Ø, I to V, and VIII) on the RSV F protein have been reported previously (20, 21). However, sites Ø, III, V, and VIII are retained only in the prefusion conformation of the F protein and recognized by the potently neutralizing antibodies 5C4 (22, 23), MPE8 (24), AM14 (25), and hRVS90 (26), respectively. Sites I, II, and IV bonding with weakly or moderately neutralizing antibodies are present on both the pre- and postfusion conformations of the F protein. Amino acid mutations in B cell epitopes on the RSV F protein are summarized in Table 2.

Amino acid substitutions with frequencies of >10% were labeled on the prefusion and postfusion conformations of F protein structures (Fig. 7). Eleven amino acid mutations occurred at the antigenic sites of the RSV A F protein. Positions with a mutation frequency greater than 10% contained K66E at site Ø, V384I at site I, N276S at site II, and V152I at site V of RSV A F. Twelve amino acid mutations occurred at the B cell epitopes of RSV B F, which included the substitutions S197N, I206M, and Q209R at site Ø and L172Q, S173L, and K191R at site V. These data showed that amino acid variations occurred at the key neutralizing sites (sites Ø and V) of the F gene, probably due to immune pressure, consistent with those previously reported (27).

## DISCUSSION

In this study, RSV strains collected from hospitalized children under 5 years old with ARIs in Xiamen during 2016 to 2019 were analyzed to improve the understanding of the molecular epidemiology features and evolutionary dynamics of RSV in China. Half of the children (52.3%) with RSV infection were infants ≤6 months old. RSV infection more frequently occurs in early infancy with an immature immune system (28). Although infants under 6 months of age can get protection from maternal transferred antibody, these antibodies decline rapidly in the next few months and protect infants only in the first 3 months of life (29, 30).

RSV shows a dynamic epidemiological pattern in which prevailing RSV groups shift from year to year. We observed that although the RSV ON1 and BA9 strains cocirculated in Xiamen during 2016 to 2019, the dominant prevalent subtype switched at a certain point in time between 2017 and 2018. The prevailing pattern "BBAA" of RSV

**TABLE 2** Amino acid substitutions of B cell epitopes on the RSV F proteins

| B cell epitope | Amino acids | RSV A | | RSV B | |
|---|---|---|---|---|---|
| | | Substitution(s) | % | Substitution(s) | % |
| Ø | 62–69, 196–209 | K66E | 99.16 | S197N | 98.21 |
| | | K68N | 0.84 | I206M | 76.79 |
| | | | | Q209R | 76.79 |
| I | 380–400 | V384I | 99.16 | L381F | 1.79 |
| II | 255–276 | N276S | 85.71 | L260F | 1.79 |
| | | S255R | 0.84 | L273F | 1.79 |
| | | P265S | 0.84 | S276N | 1.79 |
| III | 46–54, 305–310 | | | I305L | 1.79 |
| | | | | I309V | 1.79 |
| IV | 427–437 | | | | |
| V | 148–194 | V152I | 99.16 | A149S | 1.79 |
| | | S169G | 0.84 | L172Q | 98.21 |
| | | V178A | 0.84 | S173L | 98.21 |
| | | | | K191R | 76.79 |
| VIII | 163–181 | S169G | 0.84 | L172Q | 98.21 |
| | | V178A | 0.84 | S173L | 98.21 |
| P27 | 109–136 | A122T | 94.12 | P112S | 3.57 |
| | | K124N | 99.16 | K123E | 1.79 |
| | | M115I | 0.84 | | |
| | | Y117H | 0.84 | | |
| | | V127I | 0.84 | | |
| | | S130N | 0.84 | | |

was observed in this study, showing an alternating dominance between RSV A and B in Xiamen during the epidemic seasons. The alternating patterns of RSV A and B varied significantly globally. RSV A was the dominant subtype in South Africa from 1997 to 2008. Subsequently, a yearly alternating periodicity, "BABA," occurred in 2009 to 2012 (31). The epidemic pattern of RSV groups in Belgium changed from "AAB" to "AB" from 1996 to 2009 (32). The shifting in predominance from RSV A to B every 2 years, "ABBAABBA," was observed in Beijing, China, during 2007 to 2015 (33). To determine whether the alternating patterns of RSV groups in Xiamen are consistent with those in Beijing, long-term and continuative RSV surveillance should be conducted in the future.

Currently, the RSV ON1 and BA genotypes have become the predominant RSV strains at global scales (14). Similarly, the ON1 and BA9 genotypes were identified as the dominant circulating strains in Xiamen, China. Four novel mutation combinations were observed in our study. The novel substitution combinations V225A/G232R/E263V/T320A, A131T/T137I/T288I/T310I, and D213Y/L217P/A269V/L284P/A301P have never been reported before. Among them, the ON1 genotype with the specific mutation combination T113I/V131D/N178G/H258Q/H266L (41.2%) and the BA9 genotype with the novel mutation combination A131T/T137I/T288I/T310I (47.8%) were the most prevalent strains in Xiamen during the study period. The ON1 genotype with five specific amino acid substitutions, T113I/V131D/N178G/H258Q/H266L, evolved into an independent branch (lineage 3) with the Guangdong, Russia, and America strains. Although this ON1 strain has been found in Guangdong, Russia, and America, this substitution combination occurred as early as 2016 in Xiamen, which was the earliest time this substitution combination has been identified thus far. Therefore, it is very possible that ON1 strains circulating in Guangdong, Russia, and America were derived from Xiamen. In particular, more attention should be devoted to the impact of this substitution combination in Xiamen ON1 strains on disease severity. Li et al. showed that the disease severity of RSV infection correlated with the novel substitution combination (17). In addition, the BA9.2 subgenotype with the characteristic mutation combination A131T/T137I/T288I/T310I and A131T/T137I/P229L/T288I/T310I evolved into an independent clade, as

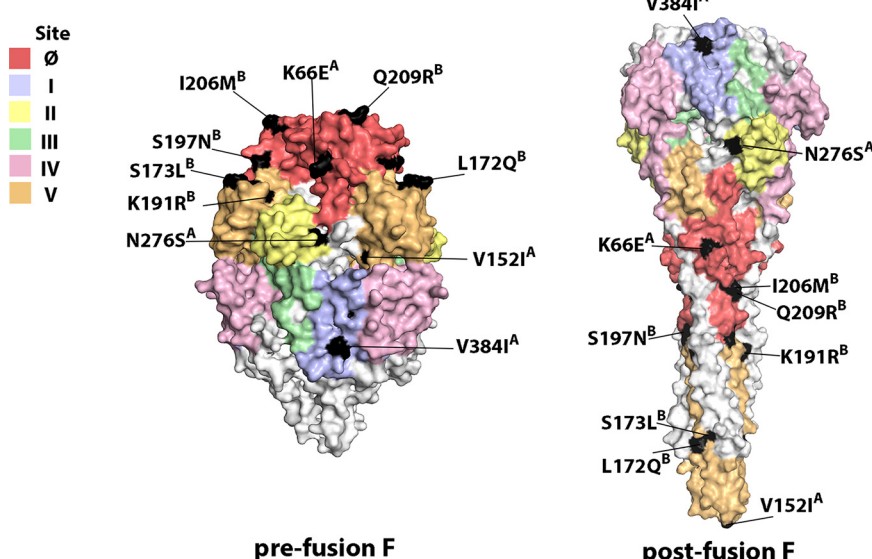

**FIG 7** Variations in neutralizing epitopes of the RSV F protein. The crystal structures of RSV prefusion and postfusion F are displayed according to the PDB files 5W23 and 3RRR, respectively. Neutralizing epitopes (Ø, I, II, III, IV, and V) were surface color coded (deep salmon, Ø; light blue, I; pale yellow, II; pale green, III; light pink, IV; light orange, V). Only amino acids with mutations of >10.0% are highlighted in black on the 3D structures of the F proteins.

well as Russia, Switzerland, and Australia strains. This novel substitution combination (A131T/T137I/P229L/T288I/T310I) probably evolved from Russia, Switzerland, Australia, and Japan strains with A131T/T137I/T288I/T310I. Xiamen is a famous travel city of Fujian Province in China, and many tourists from China and abroad visit Xiamen every year. Therefore, circulating RSV strains from different regions were probably imported into Xiamen and evolved into strains with specific local amino acid substitutions. Additionally, based on the original prevailing strains, unique substitution combinations have evolved in Xiamen. The role of all these mutations in the antigenicity of G and F proteins remains to be determined. Trento et al. have shown that despite extensive genetic diversification, the antigenic properties of GA2/ON1 viruses remain similar (34).

Similarly, continued evolution of RSV was also reflected in important antigenic sites of the RSV F protein. RSV F is the main target protein for RSV vaccine and anti-RSV antibody development (19). The prefusion F protein possessed highly neutralizing epitopes (site Ø, site III, site V, and site VIII), inducing potently neutralizing antibodies. The substitutions in our study were mainly located at sites Ø and V of RSV F; in particular, substitutions I206M/Q209R at site Ø and L172Q/S173L/K191R at site V of RSV B F were also observed in the previous results of Lu et al. (27). These mutations might affect viral antigenicity and facilitate immune escape of viruses. However, further studies are needed to determine the effects of these amino acid mutations at RSV neutralizing epitopes.

There were some limitations in our study. First, this research was confined to Xiamen, Fujian, and a single-center analysis contributed to selection bias. Second, the difference in clinical sample collection in each year might affect the seasonal distribution and prevalence estimation of RSV. Third, our sampling did not include RSV-positive milder cases from the community, only hospitalized children with severe RSV infection, which implies we may miss some significant sequence information. Additionally, a more sensitive and specific RT-PCR system is required to obtain enough RSV B G and F sequences.

**Conclusions.** This study further helps us understand the characteristics of the molecular epidemiology of RSV in China. Owing to the widespread transmission and rapid evolution of RSV ON1 and BA genotypes, long-term and continuous epidemiological surveillance is required to understand the dynamic evolution mechanism of RSV

circulating in China in the future. Additionally, the monitoring of antigenic site variations on the RSV F and G proteins will provide valuable information to improve the development of prophylactic vaccines and drugs.

## MATERIALS AND METHODS

**Sample collection.** A total of 1,026 nasopharyngeal swabs were collected from hospitalized children ≤5 years old with ARIs from Xiamen Maternal and Child Health Hospital from March 2016 to April 2019 and were primarily identified as RSV-positive specimens by direct immunofluorescence (Diagnostic Hybrids, Athens, OH, USA). This study was approved by the Ethical Committee of Xiamen Maternal and Child Health Hospital and the School of Public Health of Xiamen University. Respiratory samples were collected from patients based on the guidelines of the Ministry of Health, People's Republic of China, for public health purposes.

**Determination of RSV groups and gene sequencing.** Viral nucleic acids were extracted using a viral DNA/RNA extraction kit (GenMagBio, China) according to the manufacturer's instructions. RSV-positive samples were further confirmed by one-step quantitative real-time PCR (qRT-PCR) targeting the nucleoprotein gene. The reaction was performed in a final volume of 25 $\mu$L including 5 $\mu$L of template, 4 $\mu$L of 1 M bicine, 2.3 $\mu$L of buffer, 2 $\mu$L of 2.5 mM deoxynucleoside triphosphate (dNTP) mixture, 0.5 $\mu$L of 10 $\mu$M forward primer, 0.5 $\mu$L of 10 $\mu$M reverse primer, 0.5 $\mu$L of 10 $\mu$M probe, 0.3 $\mu$L of 1 $\mu$g/$\mu$L T4 gene 32 protein (Novoprotein, China), 0.2 $\mu$L of FastStart *Taq* DNA polymerase (Roche), 0.2 $\mu$L of avian myeloblastosis virus (AMV) reverse transcriptase (Promega, China), 0.1 $\mu$L of RNasin (Promega, China), and 9.2 $\mu$L of diethyl pyrocarbonate (DEPC)-treated water. The thermal profile was as follows: reverse transcription at 48°C for 30 min, followed by 10 min at 95°C and 45 cycles of 95°C for 15 s and 60°C for 60 s. RSV-positive samples identified by qRT-PCR were used to further determine RSV groups. The full-length F and G genes were amplified using a Qiagen OneStep RT-PCR kit (Qiagen). All sequencing was performed by Sangon Biotech Co., Ltd. (Shanghai, China). All primers and probes used in this study are listed in Table S1 in the supplemental material.

**Sequence data set construction.** For sequence data sets of the G gene, 2,714 human RSV (hRSV) complete genome sequences and 3,562 hRSV G protein full-length sequences were obtained from GenBank during July 2021. The exclusion criteria included deleting sequences collected before 2010, sequences without 24- or 20-amino acid duplications in the G gene HVR2, sequences with "NNN" regions, ambiguous nucleotides, and/or sequences with 1 to 2 nucleotide deletions or insertions causing frameshifts. Each sequence data set was aligned with MUSCLE (35). Sequences with incorrect RSV subgroup allocations were identified and added to the correct subgroup. In addition, identical nucleotide sequences were detected and removed with IQ-TREE software, and only one representative nonidentical sequence was kept (36). Finally, a total of 409 G-ectodomain sequences for RSV A and 696 for RSV B were obtained.

For sequence data sets of the F gene, 2,714 hRSV complete genome sequences and 2,876 hRSV F protein full-length sequences were downloaded from GenBank during July 2021. Curation similar to that described above was performed for sequence data sets of the F gene. Finally, a total of 829 F-ectodomain sequences for RSV A and 629 for RSV B were obtained. All selected sequences in this study are listed in the supplemental material.

**Dated phylogenetic analysis by the MCMC method.** Genotype reference strains previously reported were selected as reference sequences in our study (7, 12, 37, 38). The MCMC method was used to estimate the evolutionary rate and construct time-scaled phylogenetic trees. The GTR model was selected as the best nucleotide substitution model by ModelFinder according to the Bayesian information criterion (BIC). The data sets were analyzed by the BEAST v2.6.3 package (39) under a Relaxed Clock Exponential model. The MCMC chains for RSV A and B were run for 100 million steps to achieve convergence with sampling every 10,000 steps; convergence achievement was assessed by Tracer v1.7.1 (http://tree.bio.ed.ac.uk/software/tracer/), and effective sample size values above 200 were accepted only after 10% burn-in. The maximum credibility tree was constructed using TreeAnnotator after the first 10% of the trees were omitted as burn-in. The statistical support of the nodes was considered good when posterior probabilities were ≥0.8. The final MCMC phylogenetic tree was visualized and edited with FigTree v1.4.4 (http://tree.bio.ed.ac.uk/software/figtree/).

**Phylogenetic analysis by the ML method.** All Xiamen strains were aligned with the corresponding RSV sequence data sets using MUSCLE. GTR+F+R4 was chosen as the best-fit nucleotide substitution model with IQ-TREE software according to BIC (40). Maximum-likelihood trees were constructed using the ML method based on the GTR+F+R4 model in IQ-TREE software (41). The reliability of phylogenetic trees was assessed via 1,000 ultrafast bootstrap replicates plus an SH-like approximate likelihood ratio test (≥80% bootstrap was defined as well supported).

**Mapping amino acid substitutions in the F protein.** Three-dimensional (3D) structures of the prefusion and postfusion RSV F proteins were visualized with PyMOL, version 2.0 (Schrödinger, LLC), using two PDB files (PDB IDs: 5W23 and 3RRR). Neutralizing sites on the F protein are marked by different colors. Amino acid variations with frequencies of >10% in the antigenic sites are highlighted in black.

**Statistical analysis.** The $\chi^2$ test was applied to compare categorical variables. P values of <0.05 were considered statistically significant. Statistical analyses were conducted using SPSS, version 20 (SPSS Inc., Chicago, IL, USA). Graphs were generated using GraphPad Prism, version 6.0 (GraphPad Software Inc., San Diego, CA, USA).

**Data availability.** All F and G gene sequences of Xiamen strains have been submitted to GenBank (RSV A G, OM062619 to OM062800; RSV B G, OM062801 to OM062846; RSV A F, OM062847 to OM062964; RSV B F, OM062965 to OM063019).

## SUPPLEMENTAL MATERIAL

Supplemental material is available online only.

**SUPPLEMENTAL FILE 1**, PDF file, 0.7 MB.
**SUPPLEMENTAL FILE 2**, XLSX file, 0.1 MB.

## ACKNOWLEDGMENTS

This work was supported by the National Natural Science Foundation of China (82071783) and the Fujian Province Joint Research Project of Health Education (2019-WJ-32).

Study conception and design by Zi-Zheng Zheng, Hui-Ming Ye, and Ning-Shao Xia. Sample collection by Min Zhu and Xiao-Li Chen. Experiments performed by Yong-Peng Sun, Si-Yu Lei, Ying-Bin Wang, Hong-Sheng Qiang, Yi-Fan Yin, and Ze-Min Jiang. Data analysis by Yong-Peng Sun and Yi-Zhen Wang. Paper written by Yong-Peng Sun and Zi-Zheng Zheng. All authors critically reviewed the manuscript and approved the final version.

No potential conflict of interest was reported by us.

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
