## [Reviewer comments · Microbiology Spectrum]

Microbiology Spectrum

Molecular evolution of attachment glycoprotein (G) and fusion protein (F) genes of respiratory syncytial virus ON1 and BA9 strains in Xiamen

Yong-Peng Sun, Si-Yu Lei, Ying-Bin Wang, Yi-Zhen Wang, Hong-Sheng Qiang, Yi-Fan Yin, Ze-Min Jiang, Min Zhu, Xiao-Li Chen, Hui-Ming Ye, Zi-Zheng Zheng, and Ning-Shao Xia

Corresponding Author(s): Zi-Zheng Zheng, State Key Laboratory of Molecular Vaccinology and Molecular Diagnostics, School of Public Health, Xiamen University

Review Timeline:

Submission Date:	October 31, 2021
Editorial Decision:	December 24, 2021
Revision Received:	February 11, 2022
Editorial Decision:	February 18, 2022
Revision Received:	February 20, 2022
Accepted:	February 23, 2022

Editor: Gabriel Parra

Reviewer(s): Disclosure of reviewer identity is with reference to reviewer comments included in decision letter(s). The following individuals involved in review of your submission have agreed to reveal their identity: Monica Galiano (Reviewer #1); Adriana Delfraro (Reviewer #2)

Transaction Report:

DOI: <https://doi.org/10.1128/spectrum.02083-21>

December 24, 2021

Dr. Zi-Zheng Zheng
State Key Laboratory of Molecular Vaccinology and Molecular Diagnostics, School of Public Health, Xiamen University
422nd Siming South Road
Xiamen, Fujian 361005
China

Re: Spectrum02083-21 (Molecular evolution of attachment glycoprotein (G) and fusion protein (F) genes of Xiamen ON1 and BA9 strains)

Dear Dr. Zi-Zheng Zheng:

Thank you for submitting your manuscript to Microbiology Spectrum. Two experts on RSV evolution and diversity reviewed this manuscript. We all agree that the study has potential and could be of great interest for those working on RSV epidemiology and evolution; however, there are issues that should be addressed before being considered for publication.

I agree with both reviewers that the main issue is the lack of a strong criteria to define lineages and clusters. I could not find in the manuscript how the authors selected the viral sequences retrieved from GenBank to compare with those reported in this study. The latter is an important issue when defining lineages or reporting the presence of substitutions in proteins involved in virus protection. Please provide a rationale for the sequences selected and the list of viruses, with their corresponding genotype and cluster, used in this study as supplementary information. Note that ASM requires the submission of sequences to public repositories, so please provide the GenBank accession numbers in the text.

Link Not Available

Sincerely,

Gabriel Parra

Journals Department
Reviewer comments:

Reviewer #1 (Comments for the Author):

This manuscript describes the genetic evolution and circulation of RSV strains among hospitalised children in Xiamen during the period of 2016 to 2019. It does so by analysing phylogenetic trees with and without time-stamps. The article reads well and it is very detailed. The figures look generally clear and visually attractive. There are a variety of minor points to address, however my main concern is the lack of criteria to define lineages and clusters (details below in point 4-8). As this constitutes the basis for

most of the overall analysis, this lack of consistency invalidates most of the analysis. The authors must work out these definitions clearly before attempting to do the analyses again.

- 1) Please include the virus name "respiratory syncytial virus" in the title of the manuscript
- 2) Line 81: should say "continued" instead of "continuative"
- 3) Line 74-76: this sentence is a bit misleading. It reads as if the BA9 genotype was the first identified with a sequence repeat. Actually, all the BA genotypes are characterised by a 20aa. sequence duplication, the earliest identified in 1999 in Buenos Aires, Argentina (hence the "BA" genotype).
- 4) Figure 1: Letters in subpanels are switched. A is correct, B should be C and C should be B, according to the description on the footnote.
- 5) Figure 2: it is not clear how the lineages and clusters are defined. Materials & methods describe ML trees constructed with bootstrap values. Please include relevant bootstrap values on the tree, particularly those characterising lineages or clusters.
- 6) Please provide a definition of lineages and clusters. Is it based on bootstrap support? Presence of known references? Patristic distances? For instance, for RSV A clusters 1 and 3 of lineage 3 are monophyletic clades, whereas cluster 2 comprises one large clade and two small ones; lineage 2 cluster 1 shows a few small monophyletic clades plus some single branches, which is inconsistent. RSV B- lineage 3 -cluster 3 also include two monophyletic clades plus some single branches. These single branches are not more related to these clades within cluster 3 than to cluster 2.
- 7) How do the clusters within the Xiamen ON1 clade (Figure 1C) relate to lineages and clusters in Figure 2A? Again, clusters in Figure 1C are inconsistently defined with no explicit criteria behind. The topology shows 5 clades in this figure.
- 8) As a suggestion, there have been some recent publications with well-defined criteria to classify RSV G sequences into genotypes and/or lineages which the authors could use as guidance (Goya et al, 2019, Escalante et al, 2019, Ramaekers et al, 2020).
- 9) Also as a suggestion, a circular tree is not the best format to present the topology of RSV G sequences (see publications above as example). Also, the trees are not that large as to prevent them from showing the sequence names. The authors should include a full tree with the sequence names either as a main figure or as supplementary information, for readers to follow the correlation between the trees and downstream analysis of aminoacid signatures
- 10) Line 127 says: most of the Xiamen BA9 strains belonged to cluster 3, when it should be cluster 2.
- 11) Line 138-140: the conclusion about identical strains circulating in Xiamen during 2016-2019 is incorrect as it is not what the trees shows. The tree shows closely related sequences within e.g. RSVB L3, C2 with some possibly identical (perhaps from a single outbreak?). Because the tree does not show dates/years, we cannot see the evidence that Xiamen strains collected during successive seasons cluster together. If this is the case, then the conclusion would be that the tree shows local transmission and persistence through consecutive RSV seasons. This would have been shown much better on the time-scaled MCC trees from Figure 1. I strongly suggest the authors to reformulate this conclusion based on the Beast trees, perhaps including the timescale at the bottom of the tree 1C.
- 12) Figure 3: because of the lack of names on the ON1 sequences from figures 1 and 2, it is not possible to see the correlation between the position of these sequences on the trees compared with their characteristic amino acid substitutions.
- 13) Line 566: Figure 3D is representing the entire ectodomain of the protein G (aa. 68-320/310), not only the HVR2.
- 14) Line 171: It says T131I when it should say T113I
- 15) Figure3D looks clear and it is very informative.
- 16) Figure 4: Trees of RSV F protein: the same concerns regarding definition of lineages from point 5) apply here.
- 17) Line 227: it's aa. I206M, not I205M
- 18) Lines 228-230: The authors can't really say that "amino acid variations occurred continually (not continuedly) at the key neutralizing sites" by comparing the Xiamen F gene sequences against the reference strains. RSV A2 was isolated in 1961 and RSV B 9320 in 1977. Even though it is conserved, the F protein has evolved during these past 40-50 years. These are simply amino acid changes characterising the F protein of Xiamen strains. It would be interesting to know whether these changes were present in the other non-Xiamen sequences used to build the trees from Figure 4.
- 19) Line 303: The authors recognise the limitations of their study but missed to mention that their sampling did not include RSV positive milder cases from the community, only hospitalised children, which implies they all have had severe RSV infection. This is another limitation of the study.

Reviewer #2 (Comments for the Author):

The manuscript presents a study of the genetic variability of Respiratory Syncytial virus (RSV) in hospitalized children from a single health care center in Xiamen China. Authors studied a good number of RSV group A and B samples collected between 2016 and 2019 and complete G and F gene sequences were achieved. In the end, the study comprises 228 G sequences and 173 F sequences and their deduced aminoacidic sequences. Phylogenetic analysis and aminoacidic variability profiles were performed and discussed in terms of the molecular evolution and variability of the virus.

The study may contribute to the knowledge of the genetic variability of RSV. In addition to the classical genotyping analysis and lineage identification based on G gene, authors add an interesting analysis of the aminoacidic changes potentially involved in neutralizing -related epitopes of the F protein.

However, to improve the study and take advantage of the generated data, I have some major and minor suggestions.

My main worries are about phylogenetic analysis. It draws attention the use of a HKY model of evolution, especially for G gene. A more complex model of evolution (as GTR) may be probably more fit to the data. Using a less fit model may led to underestimate branch lengths and lower or not significant bootstrap values. I suggest using Modeltest, Findmodel or Modelgenerator (which are free) for nucleotide substitution model inference.

Regarding lineage / cluster identification. There are criteria for cluster and lineage determination, mostly based on a combination of percentage of sequence identity, monophyly of the group and a highly significant statistical support. Authors do not explain the criteria used to determine the lineages and sub-lineages described. Statistical supports are not shown. Also, authors discuss the position of samples from certain epidemic years, but we are not able to distinguish them in the tree. Perhaps dots of different color may be used or add the year of isolation for the mentioned samples. Also, criteria for choosing the sequence data base for comparison is not explicit, and this is important if origin and/or potential introduction of lineages is to be discussed.

Specific comments:

Line 41: Abstract. It's better to use the abbreviation "ARIs", instead of "ARTIs". It is the usual term to refer to Acute Respiratory Infections.

Line 49: Abstract. "Xiamen RSV strains might be derived from local persistence and external importation". I believe this sentence is somewhat speculative regarding data, as authors themselves state in discussion.

Line 66: Intro. Please rephrase. Most antibodies to G protein are poorly or partially neutralizing.

Line 67: Antigenic groups (not subtypes) have been recognized long before genetic groups or lineages, by difference in cross-neutralization assays. Later, it was demonstrated that genetic groups reproduce well and are compatible with the formerly recognized antigenic groups. Please, use group A or B throughout the text.

Line 69: When referring to HVR2 as "second hipervariable region two", please, delete "two", it is redundant. Please, correct throughout the text.

Line 74: the 60 nt duplication is not exclusive of BA9 but all BA strains. Please, correct.

Line 91 (results): Please, include here the collection years for better comprehension (2016 to 2019).

Line 104 (results) and 346 (methods): Bayesian is the tree optimization method. MCMC is the algorithm used for tree inference. Please, clarify.

Line 118 (results): Maximum likelihood (ML)

Line 131: what is an "apparent clade"?

Line 151: I suggest "share" instead of "bore", for clarity.

Line 170: this is true for the sequences analyzed, into the actual genotypes. Pleas, rephrase accordingly.

Line 201: "Phylogenetic trees with geographic annotation implied that overall, the RSV A F protein showed higher evolutionary conservation than the RSV B F protein" This claim should be discussed more in depth. Is it in accordance with previous studies? Is it influenced by the model chosen for tree reconstruction, and the diversity (in time and geographic origin) of the samples used for comparison? (and cites should be provided).

Line 211: conservation of F gene in comparison with G gene is a well-known fact for RSV, and cites should be provided.

Line 238: Discussion: "Although infants under 6 months of age derived RSV-specific antibodies from their mothers, these neutralizing antibodies declined rapidly and only protected infants in the first 3 months of life". Although the idea that you want to convey is deduced, I suggest rephrasing for better understanding.

Line 241. Please, use RSV groups instead of subtypes throughout the text.

Line 287: Maybe this sentence should be softened, as the authors themselves point out, considering their own data.

Line 312, conclusions: Please correct "continuous" instead of "continual".

Line 328 (Methodology): "Detection and distinguishing of RSV subtypes and gene sequencing. Please substitute for: "Determination of RSV groups"

Line 345: please submit the sequences and provide the accession numbers in the revised manuscript.

Line 346: As pointed out above, Bayesian is the method. MCMC is the algorithm used for tree inference. Please, clarify.

Staff Comments:

Preparing Revision Guidelines

Please return the manuscript within 60 days; if you cannot complete the modification within this time period, please contact me. If you do not wish to modify the manuscript and prefer to submit it to another journal, please notify me of your decision immediately so that the manuscript may be formally withdrawn from consideration by Microbiology Spectrum.

The manuscript presents a study of the genetic variability of Respiratory Syncytial virus (RSV) in hospitalized children from a single health care center in Xiamen China. Authors studied a good number of RSV group A and B samples collected between 2016 and 2019 and complete G and F gene sequences were achieved. In the end, the study comprises 228 G sequences and 173 F sequences and their deduced aminoacidic sequences. Phylogenetic analysis and aminoacidic variability profiles were performed and discussed in terms of the molecular evolution and variability of the virus.

The study may contribute to the knowledge of the genetic variability of RSV. In addition to the classical genotyping analysis and lineage identification based on G gene, authors add an interesting analysis of the aminoacidic changes potentially involved in neutralizing -related epitopes of the F protein.

However, to improve the study and take advantage of the generated data, I have some major and minor suggestions.

My main worries are about phylogenetic analysis. It draws attention the use of a HKY model of evolution, especially for G gene. A more complex model of evolution (as GTR) may be probably more fit to the data. Using a less fit model may led to underestimate branch lengths and lower or not significant bootstrap values. I suggest using Modeltest, Findmodel or Modelgenerator (which are free) for nucleotide substitution model inference.

Regarding lineage / cluster identification. There are criteria for cluster and lineage determination, mostly based on a combination of percentage of sequence identity, monophyly of the group and a highly significant statistical support. Authors do not explain the criteria used to determine the lineages and sub-lineages described. Statistical supports are not shown. Also, authors discuss the position of samples from certain epidemic years, but we are not able to distinguish them in the tree. Perhaps dots of different color may be used or add the year of isolation for the mentioned samples. Also, criteria for choosing the sequence data base for comparison is not explicit, and this is important if origin and/or potential introduction of lineages is to be discussed.

Specific comments:

Line 41: Abstract. It's better to use the abbreviation "ARIs", instead of "ARTIs". It is the usual term to refer to Acute Respiratory Infections.

Line 49: Abstract. "Xiamen RSV strains might be derived from local persistence and external importation". I believe this sentence is somewhat speculative regarding data, as authors themselves state in discussion.

Line 66: Intro. Please rephrase. Most antibodies to G protein are poorly or partially neutralizing.

Line 67: Antigenic groups (not subtypes) have been recognized long before genetic groups or lineages, by difference in cross-neutralization assays. Later, it was demonstrated that genetic groups reproduce well and are compatible with the formerly recognized antigenic groups. Please, use group A or B throughout the text.

Line 69: When referring to HVR2 as "second hipervariable region two", please, delete "two", it is redundant. Please, correct throughout the text.

Line 74: the 60 nt duplication is not exclusive of BA9 but all BA strains. Please, correct.

Line 91 (results): Please, include here the collection years for better comprehension (2016 to 2019).

Line 104 (results) and 346 (methods): Bayesian is the tree optimization method. MCMC is the algorithm used for tree inference. Please, clarify.

Line 118 (results): Maximum likelihood (ML)

Line 131: what is an “apparent clade”?

Line 151: I suggest “share” instead of “bore”, for clarity.

Line 170: this is true for the sequences analyzed, into the actual genotypes. Please, rephrase accordingly.

Line 201: “Phylogenetic trees with geographic annotation implied that overall, the RSV A F protein showed higher evolutionary conservation than the RSV B F protein” This claim should be discussed more in depth. Is it in accordance with previous studies? Is it influenced by the model chosen for tree reconstruction, and the diversity (in time and geographic origin) of the samples used for comparison? (and cites should be provided).

Line 211: conservation of F gene in comparison with G gene is a well-known fact for RSV, and cites should be provided.

Line 238: Discussion: “Although infants under 6 months of age derived RSV-specific antibodies from their mothers, these neutralizing antibodies declined rapidly and only protected infants in the first 3 months of life”. Although the idea that you want to convey is deduced, I suggest rephrasing for better understanding.

Line 241. Please, use RSV groups instead of subtypes throughout the text.

Line 287: Maybe this sentence should be softened, as the authors themselves point out, considering their own data.

Line 312, conclusions: Please correct “continuous” instead of “continual”.

Line 328 (Methodology): “Detection and distinguishing of RSV subtypes and gene sequencing. Please substitute for: “Determination of RSV groups”

Line 345: please submit the sequences and provide the accession numbers in the revised manuscript.

Line 346: As pointed out above, Bayesian is the method. MCMC is the algorithm used for tree inference. Please, clarify.

February 11, 2022

Microbiology Spectrum

Editorial Office

Spectrum02083-21R1: Molecular evolution of attachment glycoprotein (G) and fusion protein (F) genes of respiratory syncytial virus ON1 and BA9 strains in Xiamen

Dear Editors,

Thank you for the very professional review of our manuscript. The manuscript (**Spectrum02083-21R1**) has been revised carefully according to your constructive comments. Please find below our responses to each point raised by editors and reviewers.

We sincerely hope that this revised manuscript is suitable for publication. Thank you again for your professional handling of the manuscript and the very helpful comments.

Yours sincerely,

Zi-Zheng Zheng

Professor

State Key Laboratory of Molecular Vaccinology and Molecular Diagnostics, National Institute of Diagnostics and Vaccine Development in Infectious Diseases, School of Public Health, Xiamen University, Xiamen 361002, Fujian, PR China.

Phone: +86-13860181485, +86-0592-2880626

Fax: +86-0592-2181258

E-mail addresses: zhengzizheng@xmu.edu.cn

Responses to each point raised by reviewers and editors:

Thank you for submitting your manuscript to Microbiology Spectrum. Two experts on RSV evolution and diversity reviewed this manuscript. We all agree that the study has potential and could be of great interest for those working on RSV epidemiology and evolution; however, there are issues that should be addressed before being considered for publication.

I agree with both reviewers that the main issue is the lack of a strong criteria to define lineages and clusters. I could not find in the manuscript how the authors selected the viral sequences retrieved from GenBank to compare with those reported in this study. The latter is an important issue when defining lineages or reporting the presence of substitutions in proteins involved in virus protection. Please provide a rationale for the sequences selected and the list of viruses, with their corresponding genotype and cluster, used in this study as supplementary information. Note that ASM requires the submission of sequences to public repositories, so please provide the GenBank accession numbers in the text.

Response: Thanks for the editor's suggestions. The detail criteria to define subgenotypes and lineages could be found in Results and also included in the response to two reviewers. The rationale for the sequences selected was described in Materials and Methods (Page 13-14, Line 358-374) and the list of viruses were provided as supplementary information. The GenBank accession numbers were provided in Page 13, Line 354-356.

When submitting the revised version of your paper, please provide (1) point-by-point responses to the issues raised by the reviewers as file type "Response to Reviewers," not in your cover letter, and (2) a PDF file that indicates the changes from the original submission (by highlighting or underlining the changes) as file type "Marked Up Manuscript - For Review Only". Please use this link to submit your revised manuscript - we strongly recommend that you submit your paper within the next 60

days or reach out to me. Detailed instructions on submitting your revised paper are below.

Reviewer comments:

Reviewer #1 (Comments for the Author):

This manuscript describes the genetic evolution and circulation of RSV strains among hospitalised children in Xiamen during the period of 2016 to 2019. It does so by analysing phylogenetic trees with and without time-stamps. The article reads well and it is very detailed. The figures look generally clear and visually attractive. There are a variety of minor points to address, however my main concern is the lack of criteria to define lineages and clusters (details below in point 4-8). As this constitutes the basis for most of the overall analysis, this lack of consistency invalidates most of the analysis. The authors must work out these definitions clearly before attempting to do the analyses again.

1) Please include the virus name "respiratory syncytial virus" in the title of the manuscript

Response: Following the reviewer's advice, the virus name "respiratory syncytial virus" was added in the title of the manuscript.

2) Line 81: should say "continued" instead of "continuative"

Response: In Page 4, Line 79, "continued" replaced "continuative"

3) Line 74-76: this sentence is a bit misleading. It reads as if the BA9 genotype was the first identified with a sequence repeat. Actually, all the BA genotypes are characterised by a 20aa. sequence duplication, the earliest identified in 1999 in Buenos Aires, Argentina (hence the "BA" genotype).

Response: We thank the reviewer for this suggest. In Page 4, Line 72-73, this sentence was modified as “The BA9 genotype was first identified in Niigata, Japan, in 2006 (12). Like other BA genotypes, a sequence repeat of 20 amino acids was also inserted into HVR2 (13).”

4) Figure 1: Letters in subpanels are switched. A is correct, B should be C and C should be B, according to the description on the footnote.

Response: We apologize for the description in this section. According to comment 11, we remade Figure 1 and modified the description on the footnote in Page 19, Line 557-562.

5) Figure 2: it is not clear how the lineages and clusters are defined. Materials & methods describe ML trees constructed with bootstrap values. Please include relevant bootstrap values on the tree, particularly those characterising lineages or clusters.

Response: Thanks a lot for this suggestion. The reliability of phylogenetic trees was assessed via 1000 ultrafast bootstrap replicates plus SH-like approximate likelihood ratio test ($\geq 80\%$ bootstrap were defined as well-supported). The relevant bootstrap values characterizing lineages were labeled on the trees (Figure 2A, 2B, S1 and S2).

6) Please provide a definition of lineages and clusters. Is it based on bootstrap support? Presence of known references? Patristic distances? For instance, for RSV A clusters 1 and 3 of lineage 3 are monophyletic clades, whereas cluster 2 comprises one large clade and two small ones; lineage 2 cluster 1 shows a few small monophyletic clades plus some single branches, which is inconsistent. RSV B-lineage 3 -cluster 3 also include two monophyletic clades plus some single branches. These single branches are not more related to these clades within cluster 3 than to cluster 2.

Response: Many thanks for this comment. The subgenotypes for Xiamen ON1 and BA9 genotype and lineages within subgenotypes were defined based on ultrafast

bootstrap values ($\geq 80\%$) and patristic distances (0.01 for ON1 genotype and 0.005 for BA9 genotype), which were detailedly described in Page 6, Line 125-127. The similar approach was applied to define lineages in ML trees of the global RSV G and F protein, which were respectively described in Page 7, Line 178-179 and Page 8, Line 210-212.

7) How do the clusters within the Xiamen ON1 clade (Figure 1C) relate to lineages and clusters in Figure 2A? Again, clusters in Figure 1C are inconsistently defined with no explicit criteria behind. The topology shows 5 clades in this figure.

Response: Time-scaled MCC trees (Figure 1) were remade based on GTR model and ML trees of Xiamen ON1 and BA9 strains were constructed (Figure 2A and 2B) based on the GTR+F+R4 model. Time-scaled MCC trees and ML trees of Xiamen ON1 strains are topologically similar, both of which shows 5 clades.

8) As a suggestion, there have been some recent publications with well-defined criteria to classify RSV G sequences into genotypes and/or lineages which the authors could use as guidance (Goya et al, 2019, Escalante et al, 2019, Ramaekers et al, 2020).

Response: Thank you very much for this suggestion, which can really be a great help. The well-defined criteria to subgenotypes and lineages was identified based on three articles recommended above.

9) Also as a suggestion, a circular tree is not the best format to present the topology of RSV G sequences (see publications above as example). Also, the trees are not that large as to prevent them from showing the sequence names. The authors should include a full tree with the sequence names either as a main figure or as supplementary information, for readers to follow the correlation between the trees and downstream analysis of aminoacid signatures

Response: We followed the reviewer's suggestion. Rectangular trees are used to present the topology of Xiamen ON1 and BA9 strains (Figure 2A and 2B) and the

sequence names are shown in the trees. However, the ML trees for global RSV G (Figure 3 and S1) and F sequences (Figure S2 and S3) are too large to show the sequence names. Only circular trees for G protein of Xiamen strains and global circulating strains are used as the main figure (Figure 3) and others (Figure S1-S3) as supplementary figures. Xiamen strains are highlighted by the colored dots and the interesting sequences from other countries are labelled by the colored star (Figure 3 and S3). The list of all viruses constructing phylogenetic trees were provided as supplementary information.

10) Line 127 says: most of the Xiamen BA9 strains belonged to cluster 3, when it should be cluster 2.

Response: We apologize for this description. According to the renewed figure (Figure 3), this sentence was modified in Page 7, Line 181-182. “All the Xiamen BA9 strains (marked by colored dots) belonged to Lineage 4 (Figure 3B)”

11) Line 138-140: the conclusion about identical strains circulating in Xiamen during 2016-2019 is incorrect as it is not what the trees shows. The tree shows closely related sequences within e.g. RSVB L3, C2 with some possibly identical (perhaps from a single outbreak?). Because the tree does not show dates/years, we cannot see the evidence that Xiamen strains collected during successive seasons cluster together. If this is the case, then the conclusion would be that the tree shows local transmission and persistence through consecutive RSV seasons. This would have been shown much better on the time-scaled MCC trees from Figure 1. I strongly suggest the authors to reformulate this conclusion based on the Beast trees, perhaps including the timescale at the bottom of the tree 1C.

Response: We thank the reviewer for the comment. To show clearly the relationship of sequences in the topology, we modified Figure 1 and the names of all Xiamen strains and collection time were shown in the graph. Xiamen ON1 strains included 5 clades, one of which consisted of closely related strains collected from successive epidemic seasons (2016-2019) in ON1 lineage in the phylogenetic tree for RSV A

strains. Figure 2 illustrated that these strains in this clade included the same amino acid substitution combination (T113I/V131D/N178G/H258Q/H266L), belonging to subgenotype ON1.1. Likewise, in BA9 lineage in the phylogenetic tree for RSV B strains, we observed that Xiamen BA9 strains circulating in 2017 occurred again in 2019. Figure 2 illustrated that these strains included the same amino acid substitution combination (A131T/T137I/T288I/T310I), belonging to BA9.2 lineage. These representative sequences were labelled by red stars.

12) Figure 3: because of the lack of names on the ON1 sequences from figures 1 and 2, it is not possible to see the correlation between the position of these sequences on the trees compared with their characteristic amino acid substitutions.

Response: The names of Xiamen ON1 and BA9 sequences were shown in Figure 1 and 2. ON1 and BA9 subgenotypes were defined (Figure 2A and 2B). The representative sequences from subgenotypes were labeled by red star and their characteristic amino acid substitutions were shown in Figure 2D and 2E.

13) Line 566: Figure 3D is representing the entire ectodomain of the protein G (aa. 68-320/310), not only the HVR2.

Response: We apologize for this wrong. In Page 19, Line 564, “the G protein HVR2” was replaced by “the G protein ectodomain”.

14) Line 171: It says T131I when it should say T113I

Response: We apologize for this wrong. T131I was replaced by T113I throughout the text.

15) Figure3D looks clear and it is very informative.

Response: Yes, it is

16) Figure 4: Trees of RSV F protein: the same concerns regarding definition of lineages from point 5) apply here.

Response: The approach to defining lineages in ML trees of RSV F was similar to that of RSV G, which were described in Page 8, Line 210-211.

17) Line 227: it's aa. I206M, not I205M

Response: We apologize for this wrong. In Page 9, Line 240, I205M was replaced by I206M.

18) Line 228-230: The authors can't really say that "amino acid variations occurred continually (not continuedly) at the key neutralizing sites" by comparing the Xiamen F gene sequences against the reference strains. RSV A2 was isolated in 1961 and RSV B 9320 in 1977. Even though it is conserved, the F protein has evolved during these past 40-50 years. These are simply amino acid changes characterising the F protein of Xiamen strains. It would be interesting to know whether these changes were present in the other non-Xiamen sequences used to build the trees from Figure 4.

Response: We thank the reviewer for the comment. We rephrased this sentence as "These data showed that amino acid variations occurred at the key neutralizing sites (sites Ø and V) of the F gene due to immune pressure, consistent with those previously reported (27)." in Page 9, Line 241-243.

No Xiamen-specific amino acid changes were found in the F protein of Xiamen strains. The key substitutions, I206M/Q209R at site Ø and L172Q/S173L/K191R at site V of RSV B F, were found in other regions or countries. This section was described in Page 12, Line 306-309. "The substitutions in our study were mainly located at sites Ø and V of RSV F; in particular, substitutions I206M/Q209R at site Ø and L172Q/S173L/K191R at site V of RSV B F were also observed in the previous results of Bin Lu et al. (33)."

19) Line 303: The authors recognise the limitations of their study but missed to

mention that their sampling did not include RSV positive milder cases from the community, only hospitalised children, which implies they all have had severe RSV infection. This is another limitation of the study.

Response: Thanks for this suggestion. In Page 12, Line 315-317, we added the description about sampling limitation. “Third, our sampling did not include RSV positive milder cases from the community, only hospitalized children with severe RSV infection, which implies we may miss some significant sequence information.”

Reviewer #2 (Comments for the Author):

The manuscript presents a study of the genetic variability of Respiratory Syncytial virus (RSV) in hospitalized children from a single health care center in Xiamen China. Authors studied a good number of RSV group A and B samples collected between 2016 and 2019 and complete G and F gene sequences were achieved. In the end, the study comprises 228 G sequences and 173 F sequences and their deduced aminoacidic sequences. Phylogenetic analysis and aminoacidic variability profiles were performed and discussed in terms of the molecular evolution and variability of the virus.

The study may contribute to the knowledge of the genetic variability of RSV. In addition to the classical genotyping analysis and lineage identification based on G gene, authors add an interesting analysis of the aminoacidic changes potentially involved in neutralizing -related epitopes of the F protein.

However, to improve the study and take advantage of the generated data, I have some major and minor suggestions.

My main worries are about phylogenetic analysis. It draws attention the use of a HKY model of evolution, especially for G gene. A more complex model of evolution (as GTR) may be probably more fit to the data. Using a less fit model may led to

underestimate branch lengths and lower or not significant bootstrap values. I suggest using Modeltest, Findmodel or Modelgenerator (which are free) for nucleotide substitution model inference.

Response: Following the reviewer's advice, we used ModelFinder to choose the best-fit nucleotide substitution model according to BIC. The GTR model was selected as the best nucleotide substitution model for time-scaled phylogenetic trees. GTR+F+R4 was chosen as the best-fit nucleotide substitution model for ML trees.

Regarding lineage / cluster identification. There are criteria for cluster and lineage determination, mostly based on a combination of percentage of sequence identity, monophyly of the group and a highly significant statistical support. Authors do not explain the criteria used to determine the lineages and sub-lineages described. Statistical supports are not shown. Also, authors discuss the position of samples from certain epidemic years, but we are not able to distinguish them in the tree. Perhaps dots of different color may be used or add the year of isolation for the mentioned samples. Also, criteria for choosing the sequence data base for comparison is not explicit, and this is important if origin and/or potential introduction of lineages is to be discussed.

Response: Many thanks for this very significant comment. Another reviewer raised the same questions. The reliability of phylogenetic trees was assessed via 1000 ultrafast bootstrap replicates plus SH-like approximate likelihood ratio test ($\geq 80\%$ bootstrap were defined as well-supported). The relevant bootstrap values characterizing lineages were labeled on the trees (Figure 2A, 2B, S1 and S2). The subgenotypes for Xiamen ON1 and BA9 genotype and lineages within subgenotypes were defined based on ultrafast bootstrap values ($\geq 80\%$) and patristic distances (0.01 for ON1 genotype and 0.005 for BA9 genotype), which were detailly described in Page 6, Line 125-127. The similar approach was applied to define lineages in ML trees of the global RSV G and F protein, which were respectively described in Page 7, Line 177-179 and Page 8, Line 210-211.

Xiamen strains are highlighted by the colored dots and the interesting sequences from other countries are labelled by the colored stars (Figure 3 and S3).

The criteria for choosing the sequence data were described in Materials and Methods (Page 13-14, Line 358-374)

Specific comments:

Line 41: Abstract. It's better to use the abbreviation "ARIs", instead of "ARTIs". It is the usual term to refer to Acute Respiratory Infections.

Response: Following the reviewer's advice, "ARTIs" was replaced by "ARIs" throughout the text.

Line 49: Abstract. "Xiamen RSV strains might be derived from local persistence and external importation". I believe this sentence is somewhat speculative regarding data, as authors themselves state in discussion.

Response: We agree with the reviewer. In Page 3, Line 48-49, we modified this sentence as "we speculate that Xiamen RSV strains might be derived from local persistence and external importation."

Line 66: Intro. Please rephrase. Most antibodies to G protein are poorly or partially neutralizing.

Response: In Page 4, Line 64-65, we modified the original sentence as "RSV F and G glycoproteins are the main target antigens of neutralizing antibodies and vaccines development".

Line 67: Antigenic groups (not subtypes) have been recognized long before genetic groups or lineages, by difference in cross-neutralization assays. Later, it was demonstrated that genetic groups reproduce well and are compatible with the formerly

recognized antigenic groups. Please, use group A or B throughout the text.

Response: We agree with the reviewer, “group” was replaced by “subtype” throughout the text.

Line 69: When referring to HVR2 as "second hypervariable region two", please, delete "two", it is redundant. Please, correct throughout the text.

Response: We apologize for the repeated description in this section. In Page 4, Line 67, we removed “2”.

Line 74: the 60 nt duplication is not exclusive of BA9 but all BA strains. Please, correct.

Following the reviewer’s advice, in Page 4, Line 72-73, this sentence was modified as “The BA9 genotype was first identified in Niigata, Japan, in 2006 (12). Like other BA genotypes, a sequence repeat of 20 amino acids was also inserted into HVR2 (13).”

Line 91 (results): Please, include here the collection years for better comprehension (2016 to 2019).

Response: In Page 4, Line 87-88, we added the collection time.

Line 104 (results) and 346 (methods): Bayesian is the tree optimization method. MCMC is the algorithm used for tree inference. Please, clarify.

Response: Thanks for this comment, we modified the corresponding description in Page 5, Line 97-98 and Page 14, Line 375-377.

Line 118 (results): Maximum likelihood (ML)

Response: We modified the description in Page 5, Line 122.

“Subgenotype definition and polymorphism analysis of RSV G protein”

Line 131: what is an "apparent clade"?

Response: In Page 7, Line 182-183, we rephrased the sentence as “Xiamen ON1 and

BA9 strains were not independently clustered into a clade in the RSV A and B G trees.”

Line 151: I suggest "share" instead of "bore", for clarity.

Response: Following the reviewer’s advice, in Page 6, Line 142, "bore" was replaced by "share"

Line 170: this is true for the sequences analyzed, into the actual genotypes. Pleas, rephrase accordingly.

Response: We rephrased accordingly in Page 6-7, Line 151-157.

“Actually, we also calculated the percent of each characteristic mutation combinations and observed that ON1.1 subgenotype with T113I/V131D/N178G/H258Q/H266L (41.2%, 75/182) and BA9.2 subgenotype with A131T/T137I/T288I/T310I (47.8%, 22/46) were the predominant strains in Xiamen during the 2016-2019 RSV seasons, followed by ON1.2.1 subgenotype with K134I/T249I/E262K (19.8%, 36/182) and ON1.2.4 subgenotype with V225A/G232R/E263V/T320A (8.8%, 16/182). These results above were consistent with those Figure 3F shown.”

Line 201: "Phylogenetic trees with geographic annotation implied that overall, the RSV A F protein showed higher evolutionary conservation than the RSV B F protein" This claim should be discussed more in depth. Is it in accordance with previous studies? Is it influenced by the model chosen for tree reconstruction, and the diversity (in time and geographic origin) of the samples used for comparison? (and cites should be provided).

Response: As the reviewer said, this claim should be influenced by the model chosen for tree reconstruction, and the numbers of the samples used for comparison are not enough. In the revised manuscript, the best-fit nucleotide substitution model was chosen for ML trees with ModelFinder. In addition, all the full-length RSV genomes and F gene sequences (2010-2019) were downloaded from GenBank to enhance the diversity of the samples used for comparison. Based on the new model and sequence

datasets, we reconstructed the ML trees for RSV F. This section was rephrased in Page 8-9, Line 207-215.

Line 211: conservation of F gene in comparison with G gene is a well-known fact for RSV, and cites should be provided.

Response: The corresponding cite was added in Page 9, Line 227.

Line 238: Discussion: "Although infants under 6 months of age derived RSV-specific antibodies from their mothers, these neutralizing antibodies declined rapidly and only protected infants in the first 3 months of life". Although the idea that you want to convey is deduced, I suggest rephrasing for better understanding.

Response: Thanks for this suggestion, we modified the sentence as “Although infants under 6 months of age can get protection from maternal transferred antibody, these antibodies decline rapidly in the next few months and only protect infants in the first 3 months of life” in Page 10, Line 250-252.

Line 241. Please, use RSV groups instead of subtypes throughout the text.

Response: We followed the reviewer’s suggestion, use RSV groups instead of subtypes throughout the text.

Line 287: Maybe this sentence should be softened, as the authors themselves point out, considering their own data.

Response: According to the editor’s suggest, this sentence was modified as “It is warranted to conduct more investigations to evaluate the effect of motif mutations and variations in HVR2, especially mutations in antigenic sites, on the immune response targeting the G protein” in Page 11, Line 298-300.

Line 312, conclusions: Please correct "continuous" instead of "continual".

Response: Following the reviewer’s advice, in Page 12, Line 323, we corrected "continuous" instead of "continual".

Line 328 (Methodology): "Detection and distinguishing of RSV subtypes and gene sequencing. Please substitute for: "Determination of RSV groups"

Response: Following the reviewer's advice, we modified the sentence as "Determination of RSV groups and gene sequencing".

Line 345: please submit the sequences and provide the accession numbers in the revised manuscript.

Response: The GenBank accession numbers were provided in Page 13, Line 354-356.

Line 346: As pointed out above, Bayesian is the method. MCMC is the algorithm used for tree inference. Please, clarify.

Response: Thanks for this comment, we modified the corresponding description in Page 5, Line 97-98 and Page 14, Line 375-377.

Staff Comments:

Preparing Revision GuideLine

- Point-by-point responses to the issues raised by the reviewers in a file named "Response to Reviewers," NOT IN YOUR COVER LETTER.
- Upload a compare copy of the manuscript (without figures) as a "Marked-Up Manuscript" file.
- Each figure must be uploaded as a separate file, and any multipanel figures must be

assembled into one file.

- Manuscript: A .DOC version of the revised manuscript
- Figures: Editable, high-resolution, individual figure files are required at revision, TIFF or EPS files are preferred

For complete guideLine on revision requirements, please see the journal Submission and Review Process requirements at <https://journals.asm.org/journal/Spectrum/submission-review-process>. **Submissions of a paper that does not conform to Microbiology Spectrum guideLine will delay acceptance of your manuscript. "**

Please return the manuscript within 60 days; if you cannot complete the modification within this time period, please contact me. If you do not wish to modify the manuscript and prefer to submit it to another journal, please notify me of your decision immediately so that the manuscript may be formally withdrawn from consideration by Microbiology Spectrum.

February 18, 2022

Dr. Zi-Zheng Zheng
State Key Laboratory of Molecular Vaccinology and Molecular Diagnostics, School of Public Health, Xiamen University
422nd Siming South Road
Xiamen, Fujian 361005
China

Re: Spectrum02083-21R1 (Molecular evolution of attachment glycoprotein (G) and fusion protein (F) genes of respiratory syncytial virus ON1 and BA9 strains in Xiamen)

Dear Dr. Zi-Zheng Zheng:

Thank you for submitting your manuscript to Microbiology Spectrum. As you will see your paper is very close to acceptance. Please modify the manuscript along the lines I have recommended. As these revisions are quite minor, I expect that you should be able to turn in the revised paper in less than 30 days, if not sooner. Please find my comments below.

When submitting the revised version of your paper, please provide (1) point-by-point responses to the issues I raised in your cover letter, and (2) a PDF file that indicates the changes from the original submission (by highlighting or underlining the changes) as file type "Marked Up Manuscript - For Review Only". Please use this link to submit your revised manuscript. Detailed instructions on submitting your revised paper are below.

Link Not Available

Sincerely,

Gabriel Parra

Reviewer comments:

Lines 47-51: Please consider rephrase to "Our analyses suggest that introduction of new viruses and local evolution are shaping the diversification of RSV strains in Xiamen. This study provides new insights on the evolution and spread of the ON1 and Ba9 genotypes at local and global scale"

Lines 56-59: Please consider rephrase to "we analyzed the molecular evolution of G and F genes from RSV strains circulating in Xiamen, China. This data provides new insights on the local and global transmission and could inform the development of control measurements for RSV infections"

Lines 129-120: The separation of BA9 into two sub lineages is not convincing. Please provide better description of the patristic distance analyses used to separate BA9.1 and BA9.2. Also consider to perform Root-to-tip analyses, with each independent sublineage and the complete dataset for either BA9 and ON1, to fully demonstrate that those lineages are co-circulating (e.g. Tohma et al. J Gen Virol. 2018 Aug;99(8):1027-1035). The RTT analyses could provide important information for the dynamics of the sublineages described here. The data could be included as supplementary information.

Lines 156-157: This sentence is not needed, just cite the Figure.

Line 200: Please consider the use of "by our group"

Line 238: Please use "frequency" instead of "rate".

Line 242: Please consider using the word "probably." Authors did not perform any immunoassay to support the claim that the changes observed were due to immune pressure. While it is probable, the data to support this is missing.

Line 253: Please consider to rephrase "RSV shows a dynamic epidemiological pattern in which prevailing RSV groups shift from year to year."

Lines 294-300: Please consider to rephrase "The role of all these mutations on the antigenicity of G and F proteins remains to be determine. Trento et al. (Please cite: J Virol. 2015 Aug;89(15):7776-85) have shown that despite extensive genetic diversification, the antigenic properties of GA2/ON1 viruses remain similar."

Line 301: Please consider to omit "driven by immune pressure"

Lines 308-309: Please consider to omit "change their susceptibility to antibodies targeting these epitopes." This sentence is redundant with the rest of the paragraph.

Figure 3: This figure contains too many panels and it would not print correctly. Please split these Figure into three figures. Figure 2: Panels A/B; Supplementary Figure: Panel C; Figure 3: Panels D/E. Please check the numbers for all figures in the text.

Preparing Revision Guidelines

- point-by-point responses to the issues I raised in your cover letter
- Upload a compare copy of the manuscript (without figures) as a "Marked-Up Manuscript" file.
- Each figure must be uploaded as a separate file, and any multipanel figures must be assembled into one file.
- Manuscript: A .DOC version of the revised manuscript
- Figures: Editable, high-resolution, individual figure files are required at revision, TIFF or EPS files are preferred

Please return the manuscript within 60 days; if you cannot complete the modification within this time period, please contact me. If you do not wish to modify the manuscript and prefer to submit it to another journal, please notify me of your decision immediately so that the manuscript may be formally withdrawn from consideration by Microbiology Spectrum.

February 21, 2022

Microbiology Spectrum

Editorial Office

Spectrum02083-21R1: Molecular evolution of attachment glycoprotein (G) and fusion protein (F) genes of respiratory syncytial virus ON1 and BA9 strains in Xiamen

Dear Editors,

Thank you for the very professional review of our manuscript. The manuscript (**Spectrum02083-21R1**) has been revised carefully according to your constructive comments. Please find below our responses to each point raised by editors and reviewers.

We sincerely hope that this revised manuscript is suitable for publication. Thank you again for your professional handling of the manuscript and the very helpful comments.

Yours sincerely,

Zi-Zheng Zheng

Professor

State Key Laboratory of Molecular Vaccinology and Molecular Diagnostics, National Institute of Diagnostics and Vaccine Development in Infectious Diseases, School of Public Health, Xiamen University, Xiamen 361002, Fujian, PR China.

Phone: +86-13860181485, +86-0592-2880626

Fax: +86-0592-2181258

E-mail addresses: zhengzizheng@xmu.edu.cn

Reviewer comments:

Lines 47-51: Please consider rephrase to "Our analyses suggest that introduction of new viruses and local evolution are shaping the diversification of RSV strains in Xiamen. This study provides new insights on the evolution and spread of the ON1 and Ba9 genotypes at local and global scale"

Response: Many thanks for this suggestion. We followed the reviewer's advice and rephrased this sentence as "Our analyses suggest that introduction of new viruses and local evolution are shaping the diversification of RSV strains in Xiamen. This study provides new insights on the evolution and spread of the ON1 and BA9 genotypes at local and global scale." in Page 3, Lines 47-49.

Lines 56-59: Please consider rephrase to "we analyzed the molecular evolution of G and F genes from RSV strains circulating in Xiamen, China. This data provides new insights on the local and global transmission and could inform the development of control measurements for RSV infections"

Response: Following the reviewer's advice and rephrased this sentence as "we analyzed the molecular evolution of G and F genes from RSV strains circulating in Xiamen, China. This data provides new insights on the local and global transmission and could inform the development of control measurements for RSV infections." in Page 3, Lines 53-56.

Lines 129-120: The separation of BA9 into two sub lineages is not convincing. Please provide better description of the patristic distance analyses used to separate BA9.1 and BA9.2. Also consider to perform Root-to-tip analyses, with each independent sublineage and the complete dataset for either BA9 and ON1, to fully demonstrate that those lineages are co-circulating (e.g. Tohma et al. J Gen Virol. 2018 Aug;99(8):1027-1035). The RTT analyses could provide important information for the dynamics of the sublineages described here. The data could be included as supplementary information.

Response: We thank the reviewer for this comment. Not like ON1, there was less variability between BA9 strains. Only a lower cut-off of patristic distance (0.005) could be used to define sublineages for BA9 genotype. In addition, BA9.1 and BA9.2 sublineages included the characteristic amino acid substitution combinations respectively. Thus, BA9 was separated into two sublineages. Root-to-tip analysis was performed according to the reviewer's advice and the results were provided for review as the supplementary material.

Lines 156-157: This sentence is not needed, just cite the Figure.

Response: “These results above were agreement with those Figure 3F shown.” was modified as “Figure 4A” in Page 6, Line 153.

Line 200: Please consider the use of "by our group"

Response: In Page 8, Line 196, "our group" replaced "us".

Line 238: Please use "frequency" instead of "rate".

Response: In Page 9, Line 234, " frequency " replaced " rate ".

Line 242: Please consider using the word "probably." Authors did not perform any immunoassay to support the claim that the changes observed were due to immune pressure. While it is probable, the data to support this is missing.

Response: Following the reviewer's advice, "probably" was added in the sentence in Page 9, Line 238.

Line 253: Please consider to rephrase "RSV shows a dynamic epidemiological pattern in which prevailing RSV groups shift from year to year."

Response: We followed the reviewer's suggestion and modified this sentence as "RSV shows a dynamic epidemiological pattern in which prevailing RSV groups shift from year to year." in Page 10, Lines 249-250.

Lines 294-300: Please consider to rephrase "The role of all these mutations on the antigenicity of G and F proteins remains to be determine. Trento et al. (Please cite: J Virol. 2015 Aug;89(15):7776-85) have shown that despite extensive genetic diversification, the antigenic properties of GA2/ON1 viruses remain similar."

Response: Following the reviewer's advice, we rephrased the sentence as "The role of all these mutations on the antigenicity of G and F proteins remains to be determine. Trento et al. have shown that despite extensive genetic diversification, the antigenic properties of GA2/ON1 viruses remain similar (34)." in Page 11, Lines 292-295.

Line 301: Please consider to omit "driven by immune pressure"

Response: "driven by immune pressure" was removed in Page 11, Line 296.

Lines 308-309: Please consider to omit "change their susceptibility to antibodies targeting these epitopes." This sentence is redundant with the rest of the paragraph.

Response: "change their susceptibility to antibodies targeting these epitopes." was removed in Page 11, Line 303.

Figure 3: This figure contains too many panels and it would not print correctly. Please split these Figure into three figures. Figure 2: Panels A/B; Supplementary Figure: Panel C; Figure 3: Panels D/E. Please check the numbers for all figures in the text.

Response: We thank the reviewer for the comment. Figure 2 was split into 3 figures.

Figure 2: Panels A/B; Figure 3: Panels C/D/E; Figure 4: Panels F/G

February 23, 2022

Dr. Zi-Zheng Zheng
State Key Laboratory of Molecular Vaccinology and Molecular Diagnostics, School of Public Health, Xiamen University
422nd Siming South Road
Xiamen, Fujian 361005
China

Re: Spectrum02083-21R2 (Molecular evolution of attachment glycoprotein (G) and fusion protein (F) genes of respiratory syncytial virus ON1 and BA9 strains in Xiamen)

Dear Dr. Zi-Zheng Zheng:

Your manuscript has been accepted, and I am forwarding it to the ASM Journals Department for publication. You will be notified when your proofs are ready to be viewed.

It is my suggestion to omit this sentence "However, the hypotheses above should be supported by enough data" (Lines 291-292). The discussion already acknowledges some of the limitations of this study.

Sincerely,

Gabriel Parra
Editor, Microbiology Spectrum

Journals Department
Supplemental file 2: Accept
Supplemental file 1: Accept